



# A dynamic open-source model to investigate wake dynamics in response to wind farm flow control strategies

Marcus Becker[1], Maxime Lejeune[2], Philippe Chatelain[2], Dries Allaerts[3†], Rafael Mudafort[4], and Jan-Willem van Wingerden[1]

[1]Delft Center for Systems and Control, Delft University of Technology, Delft, Netherlands
[2]Thermodynamics and Fluid Mechanics, Institute of Mechanics, Materials and Civil Engineering, Université catholique de Louvain, Louvain, Belgium
[3]Faculty of Aerospace Engineering, Delft University of Technology, Delft, Netherlands
[4]National Renewable Energies Laboratory, Golden, CO, United States of America
[†]Deceased

**Correspondence:** Marcus Becker (marcus.becker@tudelft.nl) and Maxime Lejeune (maxime.lejeune@uclouvain.be)

**Abstract.** Wind Farm Flow Control (WFFC) is the discipline of manipulating the flow between wind turbines to achieve a farm-wide goal, like power tracking, load mitigation, or power maximization. Specifically, steady-state control approaches have shown promising results in both theory and practice for power maximization. But how are they expected to perform in a dynamically changing environment? This paper presents an open-source wake modeling framework called OFF. It allows the approximation of the performance of WFFC strategies in response to environmental changes at a low computational cost. It is rooted in previously published dynamic parametric engineering models and offers a flexible and adaptable platform to explore these models further. The presented study tests the modeling framework by investigating the performance of different wake steering controllers in a 10-turbine wind farm case study based on a subset of the Dutch wind farm Hollandse Kust Noord (HKN). The case study uses a 24-hour wind direction time series based on field data and verifies subsets of the time series in LES. The results highlight how dependent yaw travel is on the controller settings and suggest where users can strike a balance between power gains and actuator usage. They also show the structural differences and similarities between steady-state and dynamic engineering models. The comparison to LES shows what time scales the surrogate models cover and how accurately. While steady-state models capture turbine power signal dynamics up to $\approx 1/570$ Hz, the dynamic wake description can predict dynamics up to $\approx 1/360$ Hz with a better correlation and normalized root-mean-square-error. Further results show that the dynamic wake description is mainly advantageous over steady-state wake models for shorter periods ($< 20$ min). The paper also opens up the discussion about the effectiveness of wind farm flow control in a time-marching manner as opposed to a steady-state viewpoint.

## 1 Introduction

Wind energy is an essential part of the modern renewable energy mix and, thereby, part of the increasing share of energy that is covered by renewables. With this increasing share, a higher responsibility comes along. Where previously only individual turbines would contribute to the electrical grid, now numerous wind farms provided 19% of the electricity demand in the EU



in 2023 (Costanzo and Brindley, 2024). With this increased relevancy, the question arises if wind farms are used to their full extent. This could be limited, among other reasons, by unintended turbine downtime, maintenance, or non-ideal operation. Wake losses count into the latter - as front-row turbines extract kinetic energy from the wind, and they inevitably slow the flow behind them down. The turbines downstream thereby experience a lower wind speed and generate less power in response. To combat this effect, WFFC methods focus on lessening the losses induced by wakes. This is achieved by modifying the behavior of the turbines from a greedy control approach to a collaborative one.

Multiple control approaches exist to address this issue. They can be sorted by the degrees of freedom they use: (i) The blade pitch (*e.g.*. Frederik et al. (2020), Coquelet et al. (2022)) (ii) the generator torque (*e.g.* Munters and Meyers (2017) ), and (iii) the (mis-)alignment of the turbine with the flow (*e.g.* Fleming et al. (2020) or Doekemeijer et al. (2021)). Broadly speaking, (i) and (ii) change how much energy is extracted from the flow field. Applied dynamically, the blade pitch can also increase wake mixing behind the turbine, which leads to a faster wake recovery. In contrast, using (iii), the alignment of the rotor allows the controller to deflect the wake in the lateral direction. This control strategy can be used to direct the wake away from downstream turbines and is referred to as wake steering. The remainder of the paper focuses on this effect and methods to determine the effectiveness of control strategies using wake steering.

To research, test and optimize control strategies for wind farms, surrogates of the real plant are needed. This mitigates risks, lowers costs, increases flexibility, and makes the problem more accessible. Alongside wind tunnel experiments (*e.g.* , Bastankhah and Porté-Agel (2016); Hulsman et al. (2024)), simulations are the predominant form to approximate wind farm behavior. Within the world of simulations, three groups can be distinguished: high, medium, and low-fidelity simulations. High-fidelity models, such as Large-Eddy Simulation (LES), provide the most accurate approximation of the flow field (*e.g.* Chatelain et al. (2013); Churchfield et al. (2012)). This does come at an increased computational cost, which has confined their application to the verification or exploration of new phenomena not yet captured by lower-fidelity models. At the other end of the spectrum, low-fidelity simulations reduce the wake behavior to a set of simple analytical equations that are efficient to solve. This, however, means that they can only describe what they have been designed for: typically a single time-averaged snapshot of the flow field (*e.g.* Jensen (1983); Bastankhah and Porté-Agel (2016)). Low-fidelity models are therefore routinely used to, for instance, optimize the orientation of all turbines in a wind farm for the entire wind rose, to make estimates of the Annual Energy Produced (AEP), or to optimize the wind farm layout.

Growing concerns about fatigue effects on wind turbine integrity, along with the rising need for ancillary service provision, have driven recent research toward a new generation of dynamic medium-fidelity models. These models are designed to address more immediate and transient phenomena, effectively bridging the gap between high- and low-fidelity approaches. By capturing the critical dynamics of high-fidelity simulations at a fraction of the computational cost, they move beyond steady-state assumptions, unlocking new possibilities for wind farm operations. Key applications include, for example, intra-hour power production predictions for grid regulation (*e.g.* Moens et al. (2024)), as well as multi-objective wake steering strategies that optimize the power output while simultaneously mitigating the turbine's loads (*e.g.* Quick et al. (2022)).

Medium-fidelity wake models are primarily categorized by the equations they use to model flow physics, balancing computational cost with accuracy. While 2D linearized RANS methods have gathered some initial success at estimating simple wake





states, they have been shown to improperly account for wake deflection (van den Broek et al., 2022). In contrast, free-vortex methods (*e.g.*, Marichal et al. (2017), Marten (2020), or van den Broek et al. (2023b)) explicitly resolve vortex dynamics, providing deeper insights into large-scale wake behavior. This capacity, to account for phenomena such as wake deflection

and wake curling, makes free-vortex methods ideal candidates to investigate wake steering. However, the computational burden associated with these methods makes them unsuitable for large parameter spaces, such as those encountered in offshore wind farms involving dozens of turbines. Additionally, they tend to become numerically unstable for large distances and are, therefore, limited in the wake length they can describe accurately.

The Dynamic Wake Meandering (DWM) model, initially proposed by Larsen et al. (2007), also opts for a Lagrangian

parametrization of the wake, describing it as a cascade of velocity deficits without explicitly solving vortex dynamics. Since its introduction, the DWM approach has been further calibrated and validated by numerous studies comparing it against both numerical and field data (Madsen et al., 2010; Larsen et al., 2017; Jonkman et al., 2018). Building on these early successes, it has been integrated into simulation softwares such as FAST-Farm (Jonkman et al., 2017) and HAWC2FARM (Liew et al., 2023). More recently, the DWM model has been reinterpreted into a series of lighter, control-oriented wake modeling frameworks

that include FLORIDyn (Gebraad and van Wingerden, 2014; Becker et al., 2022c, b; Braunbehrens et al., 2022), OnWARDS (Lejeune et al., 2022), UFloris (Foloppe et al., 2022), and SWiPLab-WFM (Kipke and Sourkounis, 2024). A common feature of these models is that they all adopt a Lagrangian description of the flow while relying on engineering wake models to capture the wake's influence. However, though similar, these models take different paths notably regarding how they handle the ambient flow field and wake deflection. They also differ in terms of the steady-state surrogate wake model, which is generally

fixed for the presented designs. And, while steady-state models have been summarized in unified toolboxes (FLORIS (NREL, 2023), PyWake (Pedersen et al., 2023), or FOXES (Schmidt et al., 2023)), dynamic engineering models have not.

The purpose of OFF (Abbreviation based on OnWARDS, FLORIDyn and FLORIS), the dynamic wake modeling framework presented in this paper, is to provide a unified, open-source toolbox that allows for easy comparison between different implementations. Specifically, the framework aims to:

– Design and implement an interface with established steady-state models, such as FLORIS (NREL, 2023) or PyWake (Pedersen et al., 2023).

– Provide a framework for prototyping Lagrangian dynamic wake models through standardized input-output structures, facilitating the replicability of results.

– Offer accessibility through open-source code written in Python.

Such a tool shall eventually allow for benchmarking and comparisons of dynamic and steady-state wake model designs and for further exploration and development of dynamic WFFC strategies at a low computational cost (as already utilized by, *e.g.* Sterle et al. (2024); Miao et al. (2024)). Further scientific contributions of this paper are:

– An investigation into the timescales captured by steady-state wake models versus those captured by dynamic wake models, providing insights to help users make informed choices based on their specific needs.





– A verification of the presented code using LES in a neutral ABL with a ten-turbine wind farm.

– A dataset based on a total of 54 h of LES simulation with varying controller settings and changing wind directions to use for further wake model analysis and synthesis.

The following paper is split into five sections. While Section 1 introduced the context of the work, Section 2 describes the presented model and its architecture, as well as details of the implementation used to generate the results from this paper. 95 Section 3 then presents a case study where a selection of yaw-steering controllers is investigated in the presented model, followed by Section 4, where a selected range of controllers is implemented in the LES. The section goes on to compare the LES results to the results predicted by the dynamic model but also in comparison with the steady-state model. Lastly, Section 5 concludes the paper and suggests pointers for future work.

## 2 Model description

The framework called OFF is designed to run generic particle-based dynamic wind farm flow simulations using three sets of states: (i) turbine states $\mathbf{x}_\mathrm{T}$, (ii) ambient states $\mathbf{x}_\mathrm{amb}$, and (iii) observation point (OP) states $\mathbf{x}_\mathrm{OP}$. Turbine states consist out of all states necessary to describe the turbine's impact on the wake, *e.g.* the turbine yaw angle and its axial induction. The ambient states characterize the flow field, with information about wind speed, direction, and ambient turbulence intensity. The observation point states finally map the world (*i.e.* inertial) coordinate system to the wake one, thereby allowing the 105 reconstruction of a snapshot of the flow field across the wind farm. The states are then updated through three consecutive steps - prediction (Equation(1)), correction (Equation (2)), and control (Equation (3)):

$$[\mathbf{x}_\mathrm{T}(k),\ \mathbf{x}_\mathrm{amb}(k),\ \mathbf{x}_\mathrm{OP}(k)] = f_\mathrm{prediction}\left(\mathbf{x}_\mathrm{T}(k-1),\ \mathbf{x}_\mathrm{amb}(k-1),\ \mathbf{x}_\mathrm{OP}(k-1),\ \mathbf{c}\right), \tag{1}$$

$$[\mathbf{x}_\mathrm{T}(k),\ \mathbf{x}_\mathrm{amb}(k),\ \mathbf{x}_\mathrm{OP}(k)] = f_\mathrm{correction}\left(\mathbf{x}_\mathrm{T}(k),\ \mathbf{x}_\mathrm{amb}(k),\ \mathbf{x}_\mathrm{OP}(k),\ \mathbf{m}(k),\ \mathbf{c}\right), \tag{2}$$

$$\mathbf{x}_\mathrm{T}(k) = f_\mathrm{control}\left(\mathbf{x}_\mathrm{T}(k),\ \mathbf{x}_\mathrm{amb}(k),\ \mathbf{x}_\mathrm{OP}(k),\ \mathbf{m}(k),\ \mathbf{c}\right), \tag{3}$$

where $\mathbf{c}$ denotes a set of parameters, $k$ the time step, and $\mathbf{m}$ a set of measurements. The prediction step advances the model by itself: it propagates and updates the information gathered at the previous time steps. The correction step then uses the current measurements to alter the predicted states, partially reconciling them with the real-flow field. The last step finally determines the control actions the turbine takes based on the current state and measurements.

Summarizing, the OFF framework offers a prototyping environment for the development and assessment of new dynamic 115 flow modeling strategies. The update steps are kept generic, thereby allowing the user to specify its own update strategy, for instance, by switching the dynamic solver or wake model used. Figure 1 depicts the here-used version of the code that follows the FLORIDyn framework and uses FLORIS v4 as a surrogate model. The implemented update steps are further detailed in the following sections: Section 2.1 further specifies the FLORIS and FLORIDyn models used, and Section 2.2 explains how external data is fed in the simulation. Lastly, Section 2.3 introduces the control law used in this paper.





**Figure 1.** Nested software architecture used for the results presented in this paper: The OFF framework provides the interface to the wake solvers, as well as the controller. In this paper, the FLORIDyn framework is used to model the state dynamics, like the wake advection. The framework approximates the flow field at the location of each turbine and uses FLORIS to calculate measurements like effective wind speeds and power generated.

## 2.1 Prediction: Wake and Turbine modeling

The prediction step is segmented into three parts: (i) propagate the states, (ii) run the steady-state surrogate model to get turbine measurement predictions and OP advection speeds for the next time step, and (iii) retrieve information relevant to the controller. The states related to a single turbine $T$ at the $x, y, z$ location $l_{T,x}, l_{T,y}, l_{T,z}$ are propagated as follows:

$$\mathbf{x}_T(k) = \mathbf{A}_1 \mathbf{x}_T(k-1), \tag{4}$$

$$\mathbf{x}_{amb}(k) = \mathbf{A}_1 \mathbf{x}_{amb}(k-1), \tag{5}$$

$$\mathbf{x}_{OP,x}(k) = \mathbf{A}_2 [\mathbf{x}_{OP,x}(k-1) + \Delta t\, \mathbf{x}_{amb,u}(k-1)] + [l_{T,x}, 0, \ldots, 0]^T,$$

$$\mathbf{x}_{OP,y}(k) = \mathbf{A}_2 [\mathbf{x}_{OP,y}(k-1) + \Delta t\, \mathbf{x}_{amb,v}(k-1)] + [l_{T,y}, 0, \ldots, 0]^T,$$

$$\mathbf{x}_{OP,z}(k) = \mathbf{A}_2 \mathbf{x}_{OP,z}(k-1) + [l_{T,z}, 0, \ldots, 0]^T, \tag{6}$$

$$\mathbf{A}_1 = \begin{bmatrix} 1 & 0 & & \mathbf{0} \\ 1 & 0 & & \\ & & \ddots & \ddots & \\ \mathbf{0} & & 1 & 0 \end{bmatrix}, \quad \mathbf{A}_2 = \begin{bmatrix} 0 & 0 & & \mathbf{0} \\ 1 & 0 & & \\ & & \ddots & \ddots & \\ \mathbf{0} & & 1 & 0 \end{bmatrix}, \tag{7}$$

where the matrices $\mathbf{A}_1$ and $\mathbf{A}_2$ handle the state propagation. With $\mathbf{A}_1$, all states besides the first one are propagated one entry further, and the last one is disregarded. The state closest to the turbine is effectively doubled. With $\mathbf{A}_2$ the first state is not doubled but overwritten by a new input. States propagated with $\mathbf{A}_1$ do not have a new input yet *e.g.* there is no new wind speed value available at this time in the simulation cycle. Therefore, the current wind speed is kept as a prediction. The OP position states, however, do have a new input, which is the rotor center location, which is why they are propagated with $\mathbf{A}_2$. Equation (6) updates them with the turbine location $l_{T,x}, l_{T,y}, l_{T,z}$, referring to the rotor center, as a new state. In a floating turbine scenario, this could be used to induce a changing turbine and wake location due to repositioning. Note that similar, more detailed state-space descriptions can be found with Gebraad and van Wingerden (2014); Becker et al. (2022a); Foloppe et al. (2022). The code internally decomposes the wind speed and direction into its $u$ and $v$ components to avoid unexpected behavior when




switching between 360 and 0 deg. These are then used along with the time step $\Delta t$ to advance the location of the OPs through a Lagrangian update; see Equation (6). The $w$ component is ignored for simplicity. Accounting for the vertical deflection of the wake center might become necessary in some contexts, *e.g.* for simulations including terrain. However, it was not deemed necessary for the application presented here, *i.e.*, an offshore wind farm with no tilting. Note that this implementation also assumes that the OP advection speed is equal to the freestream wind speed. Alternatives are the introduction of a constant fraction of the wind speed, see for instance Ciri et al. (2017), or the use of the effective wind speed predicted by the wake model, see for instance Zong and Porté-Agel (2020). One may also decide to decouple ambient particle advection from the OP advection, thereby allowing the capture of additional wake dynamics such as wake meandering (Lejeune et al., 2022). These approaches, however, increase the computational cost of the model, as it requires the evaluation of the wake equations for every OP at every time step. Equation (5) does not include inputs as new ambient state information is introduced via the correction step; see Section 2.2. Similarly, new turbine states may be introduced in the correction or in the control step; see Section 2.3.

After the states are propagated, the wake model is evaluated to retrieve predicted measurements. This process uses the so-called Temporary Wind Farm (TWF), which provides a localized approximation of the ambient and wake conditions at a specific turbine location. More specifically, the TWF maps the current dynamic state of the simulation to the corresponding steady-state configuration at any desired position, making it interpretable by the underlying wake model, *i.e.* FLORIS. For more details, we refer to Becker et al. (2022b). A block diagram example is given in Figure 2. The graph shows the equivalent of a three-turbine wind farm where turbines T1 and T2 wake turbine T3. Turbines T1 and T2 both receive input from the wind field, add their own states, and pass them on to the first OP, which adds its own states. The set of the three state vectors is then propagated downstream. Downstream, T3 is subject to the wakes of T1 and T2. To calculate the wind speed reduction, one ghost OP is interpolated for each impacting wake. The ghost OP is based on the two closest OPs in the wake and minimizes the distance between the chain of OPs and the turbine T3. Its state is a distance-based interpolation of the two parent OPs. The state information of the ghost OPs subsequently approximates the ambient conditions and wind farm surrounding turbine T3. The TWF is then passed on to the steady-state surrogate model for evaluation. This returns predicted measurements like the effective wind speed and power generated. At each time step, a new TWF is generated for each turbine individually, which leads to $n_T$ simulations of $n_T$ turbines. The resulting computational cost will be discussed in Section 4.4. This work interfaces to the FLORIS toolbox and uses the Gauss Curl Hybrid model (Bay et al., 2023) with default settings and parameters. No parameter tuning was performed to mirror a possible "out of the box" experience. The turbine model within FLORIS is based on the $c_p(u)$ and $c_t(u)$ tables of the DTU 10 MW (Bak et al., 2013), corrected with the cosine-loss law for yaw misalignment.

## 2.2 Correction: Linking measurements and states

In this work, only ambient states are corrected. Schemes to correct the wake location exist (Braunbehrens et al., 2023; Di Cave et al., 2024) but are outside of the scope of this paper. Three ambient states are considered in the presented version of the model: wind direction, wind speed, and ambient turbulence intensity. Out of these three, only the wind direction varies in the presented simulations. By design, OFF assumes that measurements are taken at the locations of the turbines. The correction step has to alter the simulation states $\mathbf{x}_{\mathrm{amb}}$ to incorporate the new information provided. The basic assumption is made that the





**Figure 2.** Schematic of the state transportation of turbine states, ambient states, and observation point states in a three turbine example. T1 and T2 wake T3. The OPs closest to T3 in the wakes of T1 and T2 are used to create a Temporary Wind Farm (TWF) to simulate the resulting conditions for T3 in the wake model. The colored cubes indicate the states that are passed between the different elements of the software.





wind direction changes uniformly for the entire wind farm. As a result, all wind direction states are overwritten with the new measurement, which is assumed to be noise-free. Practically, this is due to the fact that the measurements used for the wind
175   direction in the experiments stem from a single location, more details will follow in Section 3.1. In an alternative setup with more measurement locations available, a sensor fusion strategy is necessary. Possible approaches to use turbine measurements to correct ambient states in the field exist; like a weighted map as done by Farrell et al. (2020), a Kalman Filter by Gebraad et al. (2015), or an Ensemble Kalman Filter as applied by Becker et al. (2022a).

## 2.3   Control: State based decision making

180   The employed controller is based on Kanev (2020) and implements a yaw steering dead-band controller that relies on a Look-up Tables (LuT) aggregated using FLORIS. Specifically, this LuT associates each wind direction to a set of optimal steering angles. In a dynamic environment, the controller now has to apply the optimized angles based on the current (estimated) ambient conditions. To this end, the controller has an estimate of the wind direction $\hat{\varphi}$, which is updated based on its own value in comparison with the measured wind direction. The estimated wind direction is then used to evaluate the LuT and provide
185   new set points. More precisely, the yaw-steering control law is formulated as follows:

$$\varphi_f(k) = f_{\text{filt}}\left(\varphi_m(k),\ \varphi_m(k-1),\ \dots,\ \varphi_m(0)\right) \tag{8}$$

$$\hat{\varphi}(k) = \begin{cases} \varphi_f(k) & \text{if } |\varphi_f(k) - \hat{\varphi}(k)| > \varphi_{\text{lim}} \text{ or } k_i \left|\sum_{l=k-1}^{\tau} \varphi_f(l) - \hat{\varphi}(k)\right| > \varphi_{\text{lim}} \\ \hat{\varphi}(k-1) & \text{otherwise} \end{cases} \tag{9}$$

$$\gamma(k) = f_{\text{LuT}}(\hat{\varphi}(k)), \tag{10}$$

where $\tau$ marks the time step of the last update of $\hat{\varphi}(k)$ to a new value. The measured wind direction at the time step $k$ and
190   its filtered version are denoted by $\varphi_m(k)$ and $\varphi_f(k)$, respectively. The control law has four elements that need to be supplied: the low-pass filter, $f_{\text{filt}}$, the dead-band width $\varphi_{\text{lim}}$, the integration coefficient $k_i$, and the LuT $f_{\text{LuT}}$. These elements determine the behavior of the wind farm, and their adequate tuning is a prerequisite to efficient wake steering. The selection of the parameters $\varphi_{\text{lim}}$ and $k_i$ is the subject of the case study presented in Section 3. The $f_{\text{filt}}$ function is omitted for simplicity, instead we assume an ideal noise-free measurement of the wind direction. The LuT is first populated using the serial-refine
195   yaw optimizer integrated into FLORIS (Fleming et al., 2022). The primary input parameters for the LuT are derived from the freestream atmospheric conditions, which are parameterized as hub-height Turbulence intensity (TI), wind speed, and wind direction. While the TI is kept constant at $6\,\%$, the wind direction is discretized into 1 deg bins, and the wind speed from $6\ \text{ms}^{-1}$ to $10\ \text{ms}^{-1}$ in $1\ \text{ms}^{-1}$ steps. The baseline controller follows the same update law, with the difference that it enforces turbine alignment with $\hat{\varphi}(k)$.



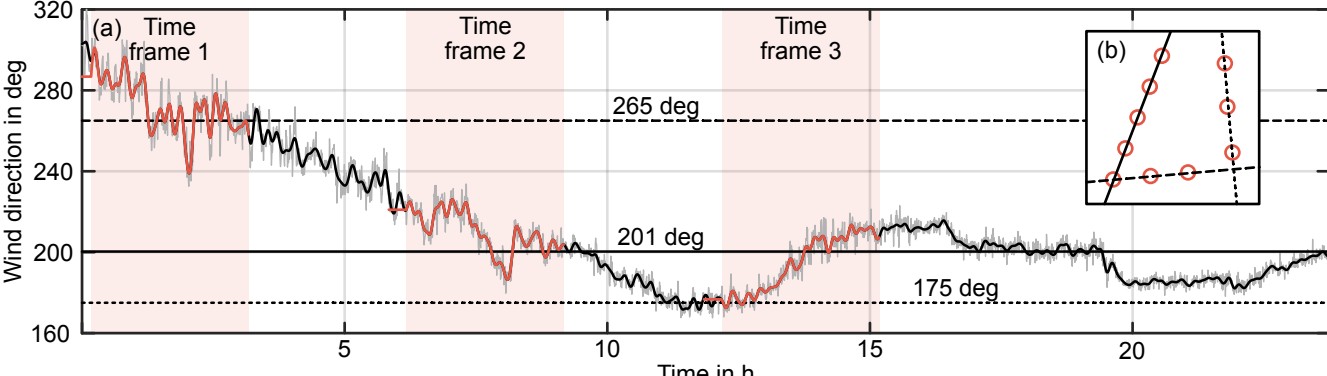

**Figure 3.** (a) The full 23 h and 45 min wind direction time series investigated in this work. The series is based on field data recorded by a vertical LiDAR at the HKN site during the 28[th] of March 2023 (Knoop, 2019), depicted in grey. The low-pass filtered data is given in black. Three marked subsets of the time series have been simulated in LES for verification purposes. Each LES Time Frame (TF) has a length of 3 h, along with a 20 min initialization period. Critical wind directions are marked in (a) and depicted relative to the farm layout in (b).

## 3 Case study

The results section is split into two parts: Section 3.1 discusses the selection and processing of the field data and the resulting simulation conditions. Section 3.2 then showcases the use of the OFF model to predict the performance of controllers and how a pre-selection can be made from a large number of controllers.

### 3.1 Simulation setup

The case study is based on the southwest corner of the HKN wind farm, which consists of ten turbines, here modeled as DTU 10 MW reference turbines with a diameter of $D = 178.3$ m (Bak et al., 2013). The layout has been scaled to preserve the same relative distances between the turbines compared to the original ones. It features three critical wind directions for which three or more turbines stand in line, namely for $\varphi \approx 175$, 201, and 265 deg. To effectively challenge the controllers, a wind direction time series that is both realistic and includes variations across all three directions (along with smooth transitions between them) is desirable. Accordingly, to drive the simulation, we use 23 h and 45 min of data recorded by a vertical ZephIR 300M wind lidar at the HKN site on March 28, 2023, as shown in Figure 3 (Knoop, 2019). This date is before the wind farm went online, which happened in December 2023[1]. The LiDAR provides horizontal and vertical wind speeds, along with wind directions, at various heights. For this study, measurements at 108 m and 133 m were used to interpolate the wind direction at a hub height of 119 m. In order to recover the underlying wind direction changes, the ensuing signal was then zero-phase low-pass filtered using a fourth-order Butterworth filter with a cut-off frequency of $1/600$ Hz, equivalent to van den Broek et al. (2023a). The filtered output was eventually fed to the yaw steering controllers, as well as the wind direction input for the precursor. For the controller, this results in an unrealistic noise-free signal, which would otherwise be a function of a filter

---

[1] www.crosswindhkn.nl, accessed 28[th] of October



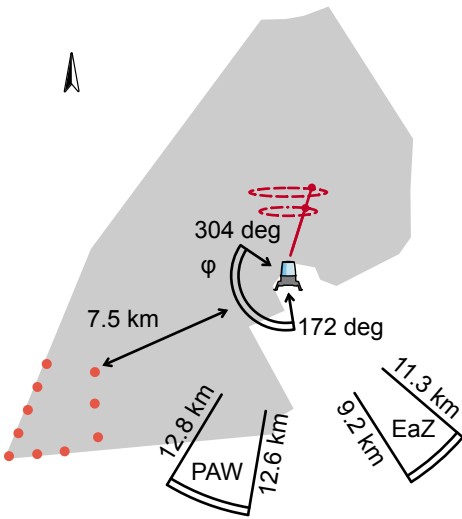

**Figure 4.** LiDAR location within the HKN wind farm site with respect to the neighboring wind farms *Prinses Amalia Windpark* (PAW) and *Egmond aan Zee* (EaZ), as well as its distance to the closest considered turbine. The measurements used in this study range from 172 deg to 304 deg, part of which, 190 to 211 deg, may be influenced by PAW. Note that the used data was recorded before HKN went online.

or distributed estimation algorithm, e.g. Annoni et al. (2019); van der Hoek et al. (2021); Howland et al. (2022). Since this work aims to demonstrate the surrogate model capabilities and not necessarily the effectiveness of an integrated wake steering

controller, the added complexity of a wind direction estimator has been left out. Figure 4 depicts the LiDAR location in the context of the HKN wind farm site and its closest neighboring wind farms[2]. The figure shows that the used wind direction range overlaps with the direction in which the Prinses Amalia Windpark is located, which may have an impact on the measurements. Therefore, for the purposes of this paper, changes in wind speed are neglected, and a constant mean wind speed of $8 \text{ ms}^{-1}$ is imposed for all simulations. This wind speed corresponds to the turbine's under-rated operation region, where the impact

of wake losses is most significant, thereby offering the greatest potential for power maximization using wake steering. The OFF simulations ran with a shear coefficient of 0.12, a turbulence intensity of 6 %, and no veer. Each turbine uses 200 OPs to describe the wake - with a time step of 5 s and a freestream wind speed of $8 \text{ ms}^{-1}$ this results in 8 km of simulated wake, or 44.9 D, which reaches beyond the boundaries of the simulated farm (approximately $5 \times 5$ km region).

### 3.2  Predicted controller performance

The controller Equation (9) updates the wind direction estimate based on either of two conditions: (i) the difference between the current wind direction and the measured direction is larger than $\varphi_{\text{lim}}$ or (ii) the integrated error exceeds the threshold. To ensure a sensible range of parameters, we investigate the balance between these two conditions: Figure 5 compares which of the two

---

[2]Adapted from map.4coffshore.com/offshorewind/, accessed 28[th] of October

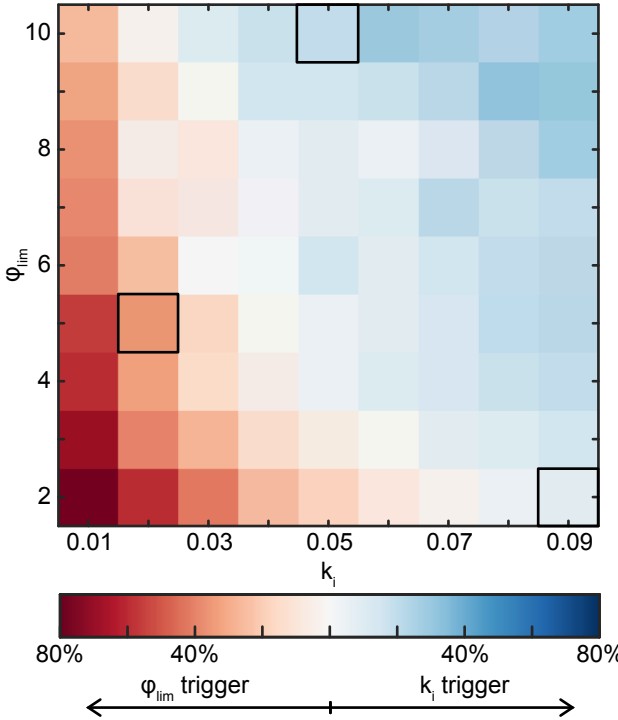

**Figure 5.** Comparison of the trigger condition that leads to an updated wind direction based on Equation (9). Red means that the controller is updated more often based on an exceeded dead band, and blue that the integrated error crosses the threshold more often. Marked squares indicate controller settings selected for verification in Section 4

triggers dominates and causes a LuT reevaluation. The results show that the chosen range of $\varphi_{\text{lim}} \in [2, 10]$ and $k_i \in [0.01, 0.09]$ leads to both cases: Either a predominant role of the threshold or one of the integration constant.

The selected ranges of $\varphi_{\text{lim}}$ and $k_i$ with a 1 deg and 0.01 discretization, respectively, lead to 81 possible combinations of dead-band settings for two types of controllers, LuT and Baseline. All 162 controllers are evaluated using OFF with the results reported in Figure 6. The figure displays the controller performance in three dimensions: (i) energy generated, (ii) number of yaw actuator activations, and (iii) accumulated yaw travel. Figure 6 (a) compares the activations with the energy generated, (b) the energy with the yaw travel, and (c) the yaw travel with the activations. All three figures are colored based on their

$\varphi_{\text{lim}}$ setting. Looking at the baseline controllers in Figure 6 (a), it becomes apparent that a smaller $\varphi_{\text{lim}}$ results in much more activations but not in an increase in energy. This is a result of a power curve that has little sensitivity to small yaw angle misalignments. On the other hand, the LuT-controlled cases still benefit from the increased number of activations, but with diminishing returns. Notably, there is little difference in the number of activations between baseline and LuT controllers. This is due to the fact that Equation (9) updates the wind direction estimates for baseline and LuT alike. In contrast, the LuT

controllers accumulate a much larger amount of yaw travel than the baseline cases, as depicted in Figure 6 (b). This is to be



**Figure 6.** (a-c) Unfolded three-dimensional performance comparison of the dead-band controllers across the full simulated time frame in OFF. Next to the energy generated by the ten-turbine wind farm, there is the accumulated yaw travel in deg and the number of times the yaw actuators are activated. The baseline controllers are colored in different shades of red, based on $\varphi_{\mathrm{lim}}$. The LuT controllers are colored in blue, respectively.

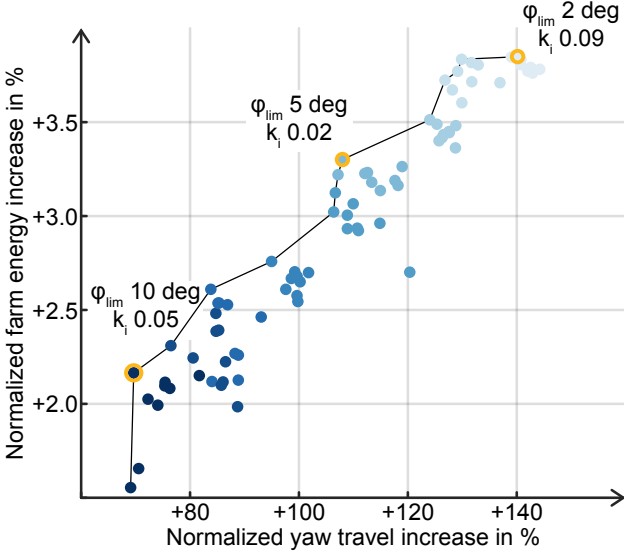

**Figure 7.** LuT controller performance normalized by the respective baseline controller with identical $\varphi_{\text{lim}}$ and $k_i$ settings. Three marked settings along the min-max Pareto front are chosen for verification. The coloring is based on $\varphi_{\text{lim}}$.

expected as the baseline controllers only drive the turbines to full alignment, while the LuT may vary between large positive and negative misalignment angles. Figure 6 (c) shows the relation between activations and yaw travel. The plot completes the picture drawn by (a) and (b): while the number of actuator activations may be similar between baseline and LuT controllers, the yaw travel is not. From these results, one could start to deduce which controllers fall within a reasonable range for set turbine limitations. For instance, if there is an average yaw activation budget of 10 times per hour per turbine, the number of relevant controllers can be reduced. In this case, $23.75 \, \text{h} \cdot 10 \, \text{turbines} \cdot 10$ activations per hour per turbine leads to a maximum of $= 2375$ activations, which limits the dead-band width at $\varphi_{\text{lim}} \geq 5 \, \text{deg}$. The results show that if yaw travel and turbine misalignment is not of concern, a LuT controller may result in a significant improvement in energy generated.

In this work, we select the controllers for verification based on the performance difference due to the switch from Baseline to LuT control. Figure 7 shows how the addition of wake steering, while maintaining the same $\varphi_{\text{lim}}$ and $k_i$, increases the amount of yaw steering in comparison to the increase in farm energy. The minimize-yaw-travel and maximize-energy Pareto front indicates several candidates that offer a trade-off between the increase in energy and the resulting increase in yaw travel. Three combinations of $\varphi_{\text{lim}}$ and $k_i$ along the front are selected for LES verification: one that yields a steep increase in energy for a relatively low increase in yaw travel ($\varphi_{\text{lim}} = 10 \, \text{deg}$ and $k_i = 0.05$), one that tries to achieve the maximum energy possible ($\varphi_{\text{lim}} = 2 \, \text{deg}$ and $k_i = 0.09$), and one intermediate configuration ($\varphi_{\text{lim}} = 5 \, \text{deg}$ and $k_i = 0.02$).

Next to the results presented in Figure 6 and 7, which summarize the overall performance, also a wind direction resolved investigation of the results can be useful. Figure 8 (a) shows the energy generated by the baseline and LuT controllers with $\varphi_{\text{lim}} = 5 \, \text{deg}$, and $k_i = 0.02$, versus the wind direction. More specifically, a sliding time window of 600 s is used to calculate the energy, as well as the mean wind direction and wind direction change. The result is a smooth transition between multiple




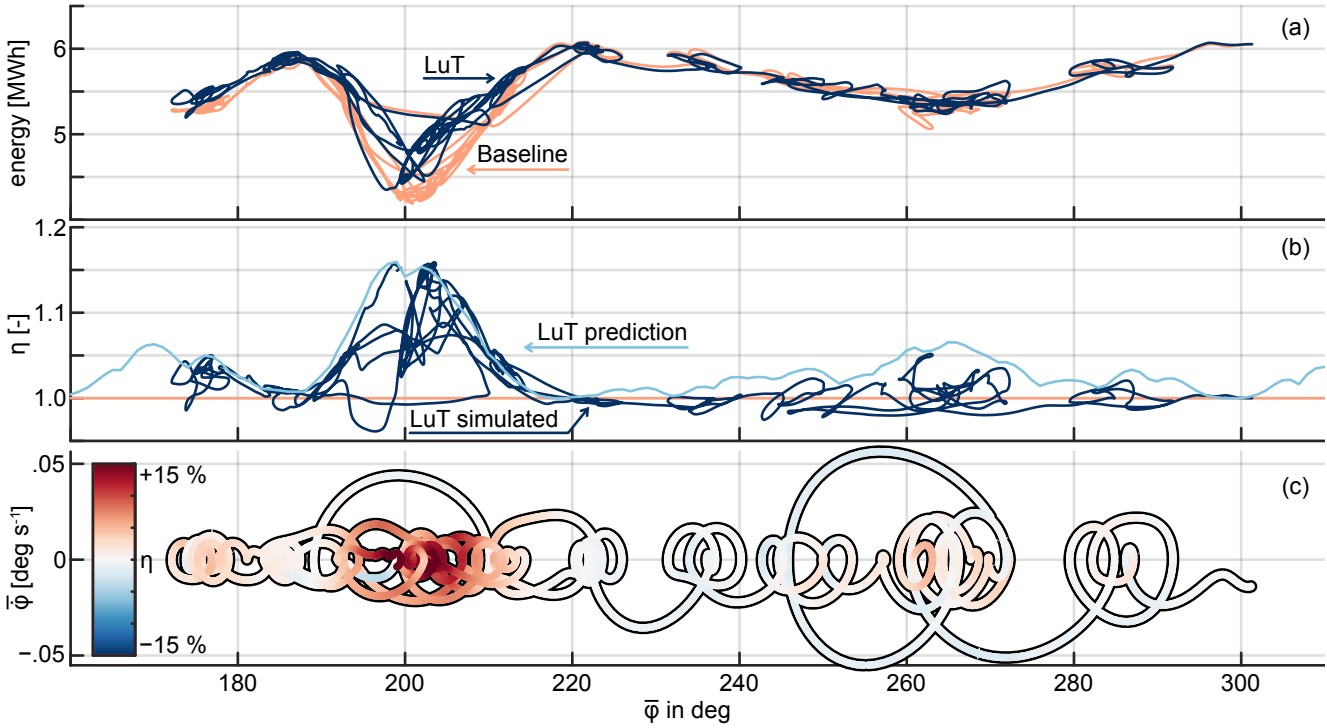

**Figure 8.** (a) Generated energy by the wind farm, calculated based on the power integrated over a sliding time window of 600 s. The energy is plotted over the mean wind direction $\bar{\varphi}$ during the 600 s for both, LuT and BL control. The resulting wind farm efficiency is given in (b) and (c). Next to the wind farm efficiency, (b) also depicts the predicted LuT steady-state wind farm efficiency. In (c), the efficiency is given as color, while the y-axis denotes the mean wind direction change $\bar{\dot{\varphi}}$ over 600 s. The controller settings are $\varphi_{\text{lim}} = 5$ deg, and $k_i = 0.02$.

10-minute average bins. The energy data is plotted over the mean wind direction and, thereby, goes back and forth along the x-axis (compare Figure 3). In direct comparison, it is evident that the LuT manages to outperform the baseline controller as expected for large parts of the wind direction, however, not for all of them. Figure 8 (b) depicts the wind farm efficiency as the ratio of the energy generated by the LuT divided by the baseline energy. The data shows that the LuT-driven controller shows advantageous behavior for wind directions between 160 deg to 220 deg but struggles to consistently outperform the baseline

in the wind direction transitions between 220 deg and 300 deg. Figure 8 (b) also depicts the wind farm efficiency as predicted by FLORIS during the LuT creation, so under ideal steady-state conditions. The difference between the achieved wind farm efficiency and the predicted one goes to show that the changing turbine states and wind direction state can lead to suboptimal performance and that the wind farm efficiency predicted by the LuT is indeed an upper limit. Lastly, Figure 8 (c) shows the wind farm efficiency over the mean wind direction, as well as the mean wind direction change. This serves as an approximated

state-space representation of the wind direction and how it influences the wind farm performance. Since the y-axis depicts the wind direction change, the state of the wind direction moves left in the lower half of the plot and right in the upper half. In conclusion, the performance of a wake steering controller is not trivial to assess in a time-marching simulation due to changes



in the flow field and in the turbine state. As a result, the wind farm can exhibit very different performance for the same wind direction and wind speed.

## 4 HKN Cases simulated in LES

This section verifies the selected controllers from Section 3.2 across the three subsets of the 24-hour period simulated in OFF and FLORIS. The OFF results are compared to both the LES and to FLORIS, allowing to investigate the effect of the added dynamics. While the following Section 4.1 further introduces the LES setup and the three time-frames, Section 4.2 investigates the power generated on a turbine, farm, and statistical level. This is followed by Section 4.3, where the energy generated is compared between the simulations.

### 4.1 Large Eddy Simulation

The ten-turbine wind farm is simulated as actuator discs in a $5 \times 5 \times 1$ km simulation domain in SOWFA (Churchfield et al., 2012). The domain is discretized in $300 \times 300 \times 100$ cells and simulated with a time step of $0.5$ s. A grid resolution of $16.6 \times 16.6 \times 10$ m was chosen to balance computational cost and accuracy. Given the turbine rotor diameter of $178.3$ m, this results in a normalized cell width of $\Delta x = \Delta y = 0.094$ D, but since the turbines are often diagonally oriented in the domain during the simulation, a worst-case ratio of $\sqrt{2}\Delta x = 0.132$ D. The neutral turbulent precursor is developed over $3 \cdot 10^4$ s. A surface roughness of $0.0002$ m enforces a horizontal turbulence intensity of $\approx 6.2$ % at hub height. The initial wind direction is kept constant at 225 deg during the precursor to allow changes of $\pm 45$ deg in the successor phase, using the same South and West inflow planes. Three 3 h successor phases are simulated in LES, as marked in Figure 3. A 1200 s spin-up phase with fixed wind direction is first run in order to fully propagate the wake, after which 10800 s of the low-pass filtered field data is used to uniformly change the wind direction. All three time series are offset to start with 225 deg, while the wind farm layout is rotated in the LES thereby ensuring the same precursor can be used accross all three simulations. The veer of the precursor is $< 2$ deg across the rotor plane, and the shear exponent is $\approx 0.075$. Figure 9 shows the wind farm in the rotated domain and a qualitative visualization of the wind directions during the simulation. The latter is achieved by a pizza-shaped histogram with bins of 2.5 deg width, translated onto the position of each turbine. Darker bins indicate more frequent wind directions, lighter ones less frequent ones, thereby visualizing the wind turbine interactions. Next to the domain orientations, Figures 9 (a-c) also depict information relevant to all three TF; (a) lists the turbine indexes, (b) the simulated domain size, and (c) the normalized distance between turbine T0 and the other turbines.

To link the dynamics back to the layout, time is also given in convective time scales. This denotes the time taken by a particle to travel a characteristic length within the domain. We choose this length to be 5 turbine diameters, as this is closely related to the spacing of the turbines; see Figure 9(c). The freestream velocity is used to normalize the characteristic length:

$$t_c = \frac{5 \cdot 178.3 \text{ m}}{8 \text{ ms}^{-1}} = 111.4 \text{ s}. \tag{11}$$

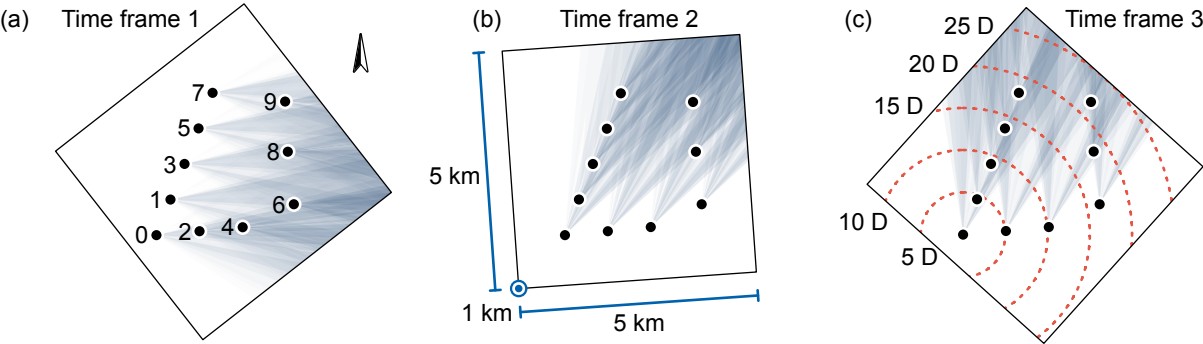

**Figure 9.** Collection of the three simulated LES TFs of the ten-turbine subset of the HKN wind farm. (a-c) feature pizza-shaped histograms of the wind direction centered in the turbine locations: darker colors indicate more frequent wind directions and, therefore, turbine interactions that happen more frequently during the TF. Additionally, (a) depicts the turbine indexes, (b) the simulated domain size, and (c) the relative distance between turbine T0 and the other turbines, normalized by turbine diameters. The domains are rotated such that the initial wind direction is aligned with the precursor, and the remaining wind direction time series can be simulated with the same inflow planes.

## 4.2 Power generated

The power generated by SOWFA is calculated based on an actuator disc model. Simulated on coarse grid, these tend to overestimate the power generated by the turbines, which is a known issue (Martinez et al., 2012; Shapiro et al., 2019). The resulting mean ratio between the power generated in SOWFA and OFF is $1.34$. Based on this mismatch, the power measurements by SOWFA in the following plots are either normalized or marked with a $c$, which denotes that the power was divided by the correction factor. Next to the LES data, the zero-phase filtered power output data from the LES is also used to analyze model and controller performance. This filtering removes the influence of turbulence on turbine power, isolating the underlying trends more consistently with the wake dynamics that OFF aims to describe. To this end, a $4^{\text{th}}$-order Butterworth filter is used with a cutoff frequency of $1/370$ Hz. The cutoff frequency is motivated by the results presented later in Figure 13 (b). Note that the individual turbine signals are filtered. Derivatives, like farm power or energy, then use either the filtered turbine power or the original signal and are marked with *lpf* if they use the filtered data.

The match between OFF and SOWFA is investigated in three ways: (i) on a selected turbine level for a selected controller, (ii) on a farm level for a selected controller, and (iii) on a statistical level. Figure 10 and 11 investigate the data collected for turbine T3. The data was recorded using the dead-band LuT and baseline controllers with $\varphi_{\text{lim}} = 5$ deg, and $k_i = 0.02$, one of the settings selected for validation based on the results in Figure 7. Turbine T3 is selected as it acts as an upstream turbine in TF 1, see Figure 9, and as a downstream turbine in TF 2 and 3. This is mirrored in Figure 10, where the turbine produces its maximum power during the initial hours of the time series. The LuT-controlled case diverges as the turbine engages in yaw steering and sacrifices power to redirect its wake. During later periods of the simulation, T3 becomes a downstream turbine




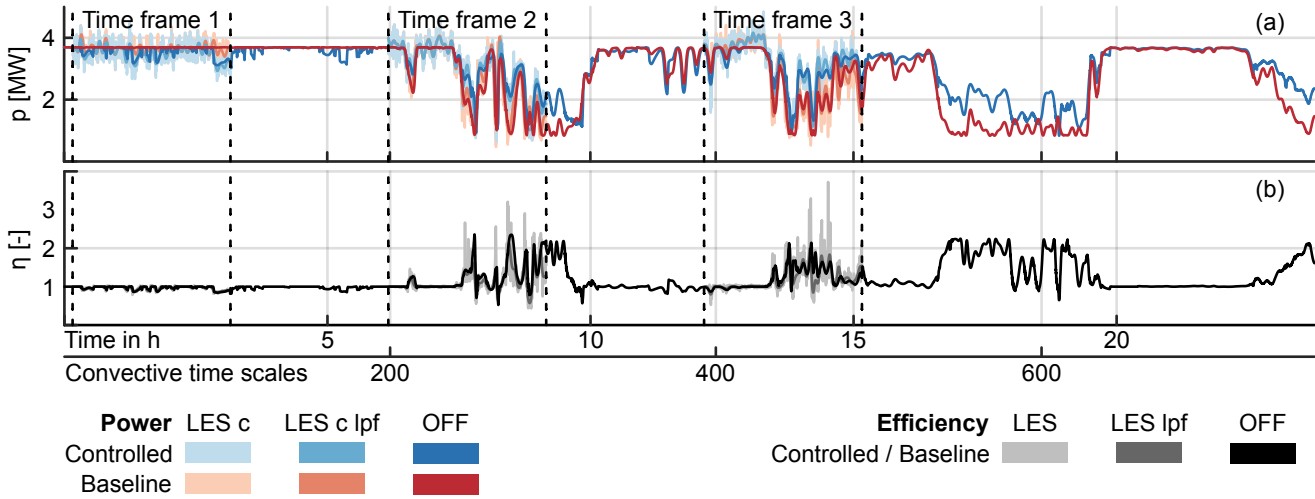

**Figure 10.** Power generated (a) and efficiency with respect to the baseline (b) of turbine T3 throughout the full simulated wind direction time series. "LES c" refers to the corrected SOWFA data, and "lpf" refers to the zero-phase low-pass filtered data. The controller settings are $\varphi_{\text{lim}} = 5$ deg, and $k_i = 0.02$. The detailed data from the time frames is provided in Figure 11.

and its power generated significantly decreases. Here, we can see an inverse effect, where T3 benefits from the yaw steering of other turbines and generates more power in the controlled case than in the baseline case.

Zooming in on the TFs simulated in LES, Figure 11 gives a more detailed look into the match of the LES data and the OFF data. Qualitatively we observe an overall fitting trend between the LES signal and the power predicted by OFF. An immediate

difference between the two is the influence of turbulence on the LES signal. This causes noticeable variations that OFF cannot predict. The low-pass filtered signal removes this discrepancy partially and shows a signal that is overall better aligned with the OFF signal. One aspect that gets lost due to this filtering is the response of the turbine power to yaw angle changes: Figure 11(b) shows the efficiency of the turbine during a period where turbine T3 engages in yaw steering to lessen the wake interaction with a downstream turbine. In OFF, the rotor misalignment causes sharp de- and increases in efficiency, while the

change is either smoothed out by filtering or hidden in the noise for the LES data. Reoccurring discrepancies between OFF and the low-pass filtered LES data show in the form of a phase shift, mainly visible with the baseline power signal: OFF displays slightly delayed reductions and recoveries compared to SOWFA. This might be the product of a too-slow advected wake, which is notable as similar models specifically slowed their advection speed down for a better match with reference data. Another difference between OFF and the LES data is visible in the turbine efficiency displayed in Figure 11(d) and (f): OFF tends to

either match or overestimate the effect of yaw steering on the turbine efficiency, compared to the filtered LES signal.

Figure 12 moves from the turbine power described previously to the farm level. As the scale increases, the differences between the signals decrease. On a farm level OFF shows a qualitatively better match than on a turbine scale, where differences become much more clear. The farm power efficiency is also more balanced compared to the turbine level; both over and

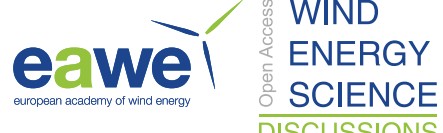

**Figure 11.** Power generated (a,c,e) and efficiency with respect to the baseline (b,d,f) of turbine T3 during the three simulated TF. "LES c" refers to the corrected SOWFA data, and "lpf" refers to the zero-phase low-pass filtered data. The data is a subset of Figure 10.



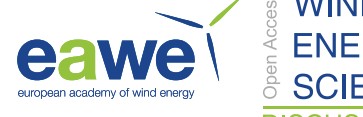

**Figure 12.** Farm power generated (a,c,d) and efficiency with respect to the baseline (b,d,f) during the three TFs. "LES c" refers to the corrected SOWFA data, and "lpf" refers to the zero-phase low-pass filtered data. The dead-band controller settings are $\varphi_{\text{lim}} = 5$ deg, and $k_i = 0.02$.





underestimations are present if there is a mismatch. Which suggests a lower bias. The data presented in Figure 12(a) and (b)
also highlights TF 1 as a difficult period for wake steering to achieve consistent gains.

The results presented in Figure 10 - 12 show the similarities but also the discrepancies between OFF and the LES with
respect to power generated. In the OFF environment, fluctuations in the power signal are due to (i) wind direction changes, (ii)
control set point changes, and (iii) delayed wake dynamics. By contrast, the LES environment also reflects fluctuations due to
turbulence and wake meandering. These latter two factors contribute to higher-frequency effects, raising the question: Which
frequency ranges does OFF effectively capture? And which frequencies could also be represented in a steady-state model?

To answer this question, we investigate the correlation between the power signals. Assuming that the discrepancies between
OFF and the LES are of a high-frequency nature, one would expect that the correlation between the two models increases as
high-frequency fluctuations are filtered out. In turn, with too aggressive filtering, the correlation should eventually decrease as
the LES signals lose components described by OFF. Based on these assumptions, the turbine individual data of TF 1-3 for the
baseline and LuT dead-band ($\varphi_{\lim} = 5$ deg, $k_i = 0.02$) controllers is correlated between OFF and the LES. A total of 180 h of
data, or 18 h per turbine, are subsequently processed. Figure 13 (a) illustrates the influence of the cutoff frequency of the $4^{\text{th}}$
order Butterworth filter applied to the LES on the correlation score recorded by OFF while Figure 13 (b) depicts the resulting
average correlation error. The average correlation error is defined as the mean distance of the turbines to 1 for all three TFs:

$$e_{\text{corr}} = \frac{1}{n_{\text{DT}}} \sum_{i_{TF}} \sum_{i_T} [1 - \text{corr}(p_{\text{OFF}}, p_{\text{LES}})], \tag{12}$$

where $p$ is the power of turbine $i_T$ in TF $i_{TF}$, and $n_{\text{DT}} = 5 + 6 + 6 = 17$ is the total number of downstream turbines considered
summed across all three TFs. Combining the baseline and controlled cases, the minimum for $e_{\text{corr}}$ is achieved for $f_{\text{cutoff}} =$
$1/370 \text{ s}^{-1} = 1/3.33 \, t_c = 0.0027 \text{ Hz}$ . In contrast to OFF, the collective minimum for FLORIS is reached at $1/520 \text{ Hz} =$
$1/5.11 \, t_c$, so at a lower frequency. This gap is explained by the added wake dynamics in OFF, as OFF uses the same FLORIS
model in its core. Additionally, we note that OFF leads to a lower error than FLORIS; while OFF finds its minimum at
$e_{\text{corr}} = 0.11$, FLORIS returns $e_{\text{corr}} = 0.19$. It should be noted that the filtering timescale used to preprocess the wind direction
signal ($1/600$ Hz) may limit OFF's performance, as it filters out relevant dynamic scales. Rerunning the LES with a higher
cutoff frequency would likely increase OFF's effective cutoff frequency estimation; however, this was not feasible within the
scope of the present work.

Figure 14 provides more insight into the source of the correlation error. Figure 14 (a,b) show the correlation error of OFF,
split into LuT cases (a) and BL cases (b). This is accompanied by the results for FLORIS, depicted in (c) and (d), also split
into LuT cases and BL cases, respectively. Upstream turbines, like T0, T1, T3, T5, and T7 for TF 1, are neglected in Figure 13
and 14 as they are operating at close-to maximum power in OFF and FLORIS, while their LES counterparts are affected by
turbulence, see for instance Figure 11 (a). As a result, the turbines modeled in OFF and FLORIS experience no excitation,
while the LES ones do. This leads to effectively no correlation between the signals.
Looking at which turbines lead to the larger $e_{\text{corr}}$ for FLORIS, the turbines in TF 1 contribute a large share, as well as turbine
T9 in TF 2. Based on Figure 9, we can see that TF 1 features long-distance turbine-to-turbine interactions. This fact, paired
with the varying wind direction, leads to a situation where the steady-state approximation of FLORIS fails and where wake



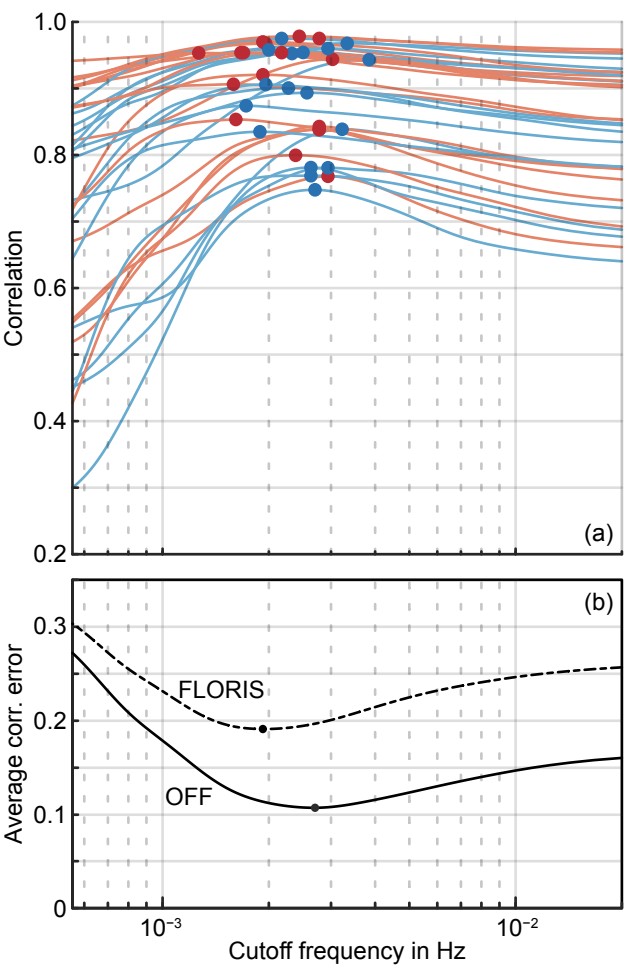

**Figure 13.** (a) Correlation of the downstream turbine power in OFF and the LES. The LES data is zero-phase low-pass filtered with varying cutoff frequencies. Blue lines are the controlled cases, red are the baseline cases. Each dot represents the maximum correlation from a given turbine. The average error is depicted in (b) and is minimal for $f_{\mathrm{cutoff}} = 1/370 = 0.0027$ Hz. Alongside, there is the line for the correlation of the FLORIS data with the LES. Its minimum is located at $1/520$ Hz. The dead-band controller settings are $\varphi_{\mathrm{lim}} = 5$ deg and $k_i = 0.02$.



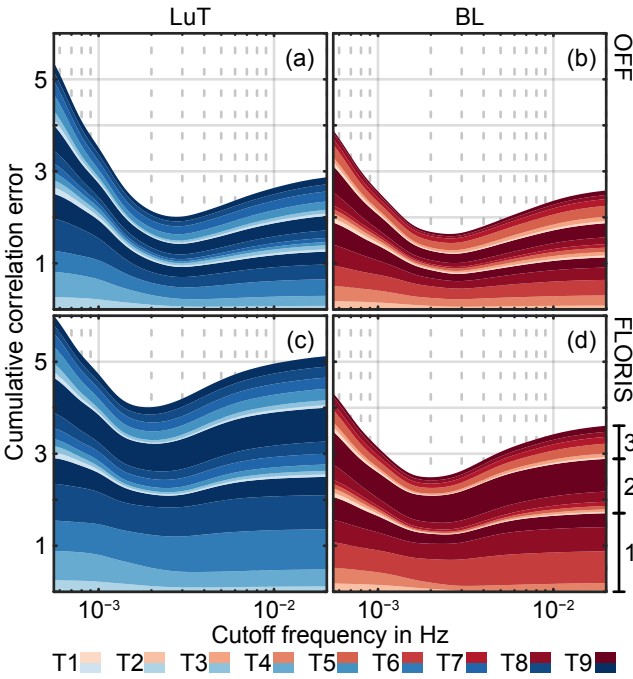

**Figure 14.** Cumulative correlation error between the turbine power from the LES and OFF (a,b) and FLORIS (c,d). The data is split into the LuT cases (a,c) and the baseline cases (b,d). The shaded areas indicate the contribution of each downstream turbine across the three TF on top of each other. With (d) it is indicated which layer relates to the corresponding TF. The dead-band controller settings are $\varphi_{\text{lim}} = 5$ deg, and $k_i = 0.02$.

dynamics play a significant role in the power generated. This also complements the observation from Figure 12 (b), where it was visible that TF 1 is a challenging case for the steady-state-based LuT controller. A notable similarity between OFF and FLORIS is that the LuT cases lead to a higher error than the baseline cases. One reason for this discrepancy could be that the turbine model does not accurately capture the impact of larger misalignment angles. This would motivate turbine model corrections as suggested by Heck et al. (2023). Additionally, this error may be partially rooted in the wake dynamics triggered by LuT control. Indeed, LuT-based wake steering tends to amplify changes in wind direction: a variation of just a few degrees in the wind direction may, under certain circumstances, induce a yaw-offset angle change that is ten times greater than the original wind direction change (Lejeune et al., 2024). This results in more frequent and larger variations in wake states.

Up to this point, Section 4.2 investigated first the turbine power, then the farm power, as well as the role of time scales. This discussion was limited to one set of controller settings $\varphi_{\text{lim}} = 5$ deg, and $k_i = 0.02$. For brevity, we denote the controller settings as $(\varphi_{\text{lim}}, k_i)$ in the following paragraph. Two more sets of settings were simulated in LES, namely $(2, 0.09)$ and $(10, 0.05)$. Table 1 summarizes characteristic error quantities for all controllers. The table combines the three TFs for each controller setting to calculate the difference between the OFF prediction and FLORIS prediction. The table lists the normalized RMSE for the turbine and farm power, as well as the correlation of both signals. The normalization was done with the corrected LES





| Model | $\varphi_{\mathrm{lim}}$ | $k_i$ | Mode | T. NRMSE [-] | T. Corr. [-] | F. NRMSE [-] | F. Corr. [-] | $f_{\mathrm{cutoff}}$ [Hz] | $e_{\mathrm{corr}}$ [-] |
|-------|------|------|------|-------|-------|--------|-------|---------|-------|
| OFF | 2 | 0.09 | LuT | 0.19 | 0.81 | 0.047 | 0.88 | 1/360 | 0.14 |
| FLORIS | 2 | 0.09 | LuT | 0.27 | 0.74 | 0.064 | 0.81 | 1/540 | 0.26 |
| OFF | 2 | 0.09 | BL | 0.20 | 0.88 | 0.048 | 0.90 | 1/430 | 0.10 |
| FLORIS | 2 | 0.09 | BL | 0.19 | 0.87 | 0.045 | 0.92 | 1/520 | 0.13 |
| OFF | 5 | 0.02 | LuT | 0.18 | 0.83 | 0.043 | 0.90 | 1/360 | 0.12 |
| FLORIS | 5 | 0.02 | LuT | 0.24 | 0.76 | 0.056 | 0.84 | 1/520 | 0.24 |
| OFF | 5 | 0.02 | BL | 0.20 | 0.88 | 0.047 | 0.91 | 1/390 | 0.10 |
| FLORIS | 5 | 0.02 | BL | 0.20 | 0.87 | 0.045 | 0.92 | 1/520 | 0.15 |
| OFF | 10 | 0.05 | LuT | 0.18 | 0.85 | 0.042 | 0.91 | 1/370 | 0.11 |
| FLORIS | 10 | 0.05 | LuT | 0.24 | 0.80 | 0.053 | 0.85 | 1/510 | 0.21 |
| OFF | 10 | 0.05 | BL | 0.20 | 0.88 | 0.048 | 0.91 | 1/370 | 0.09 |
| FLORIS | 10 | 0.05 | BL | 0.21 | 0.86 | 0.047 | 0.91 | 1/500 | 0.16 |

**Table 1.** Power error statistics for each controller tested in OFF, FLORIS, and LES. From left: T.NRMSE: the normalized root-mean-squared error calculated with the corrected turbine power LES data, T.Corr.: the correlation with the unfiltered turbine power LES signal, F.NRMSE: the normalized root-mean-squared error calculated with the corrected farm power LES data, F.Corr.: the correlation with the unfiltered farm power LES signal, $f_{\mathrm{cutoff}}$: the cutoff frequency for LES filtering, and $e_{\mathrm{corr}}$: the average correlation error.

data. The values show that the addition of dynamics renders OFF more robust towards the addition of yaw steering, compared to FLORIS: While the turbine and farm NRMSE slightly decrease for OFF, there is a notable increase for FLORIS related to the switch from BL to LuT operation. Similarly, the correlation of the farm and turbine power decreases for both OFF and
FLORIS, but the steady-state approximation results in a larger decrease, e.g., for $(2, 0.09)$, the farm power correlation by OFF decreases by $\approx -0.03$ compared to $\approx -0.11$ for FLORIS. However, both OFF and FLORIS achieve similar correlation and error results for baseline operation. An explanation can be that the LuT creates wind farm states that are more sensitive to environmental changes. As a result, the modeled wake dynamics become more relevant. Also notable is the NRMSE decrease for both models with the switch from turbine level to farm level, from values between $0.17$ and $0.27$ to values between $0.04$
and $0.06$. Consequently, model inaccuracies on a turbine level do not necessarily lead to equally large errors on a farm level. This also indicates that going forward, improved model descriptions might lead to less uncertainty on a turbine basis but might show diminishing returns on a farm scale.

### 4.3 Energy generated

The previous Section 4.2 investigated the power generated by the wind farm at different time and turbine scales. This section
complements the results with a discussion about the energy generated. More specifically, the efficiency of the wind farm is compared between the LES and the surrogate models. The efficiency is calculated as the ratio of the farm energy generated





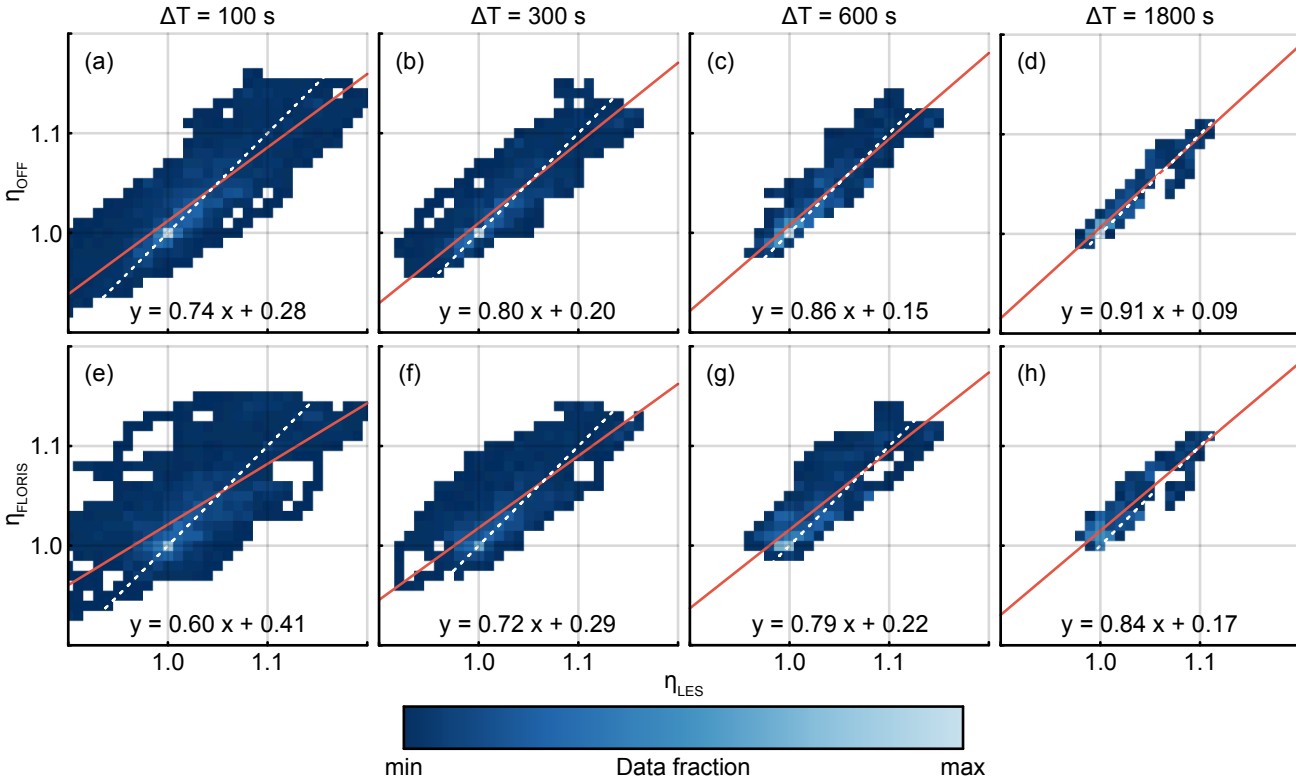

**Figure 15.** Wind farm efficiency as predicted by the surrogate models OFF (a-d) and FLORIS (e-h) and as simulated in the LES. The efficiency is calculated based on the ratio of energy generated over a time window $\Delta T$, which is equal for each column of the figure, e.g. (a) and (e). The dotted white line indicates a perfect fit, which is complimented by the linear regression of the data, given as red line and equation. The color map is normalized by the largest bin count based on the given time window. The darkest color is reserved for the smallest non-zero bin count; empty bins are not filled. Note that the distribution of $\Delta T$ is not equidistant.

using LuT control, normalized by BL control, integrated over a time window $\Delta T$:

$$\eta(t, \Delta T) = \frac{\int_t^{t+\Delta T} \Delta t \sum_{n_T} p_{\text{LuT}}(\tau) \, d\tau}{\int_t^{t+\Delta T} \Delta t \sum_{n_T} p_{\text{BL}}(\tau) \, d\tau}, \tag{13}$$

where $p$ refers to the power generated by a turbine, $\Delta t$ is the time step, and $t$ is the time. Figure 15 compares $\eta_{\text{LES}}(t, \Delta T)$, the

wind farm efficiency simulated in the LES, with $\eta_{\text{OFF}}(t, \Delta T)$ and $\eta_{\text{FLORIS}}(t, \Delta T)$, the values OFF and FLORIS predict respectively. This is done for four values of $\Delta T$ between 100 s and 1800 s with data from all three TFs, and based on the $\varphi_{\text{lim}} = 5$ deg, $k_i = 0.02$ controllers. A first observation is that the range of values for the farm efficiency decreases with increasing length of $\Delta T$. This shows the increasing convergence towards a more consistent controller performance over a longer time as well as a diminishing influence of effects at a small time scale. In comparison between OFF and FLORIS, OFF generally predicts a

narrower fit for small values of $\Delta T$, closer to the ideal correlation line. With increasing $\Delta T$, this difference diminishes, and the

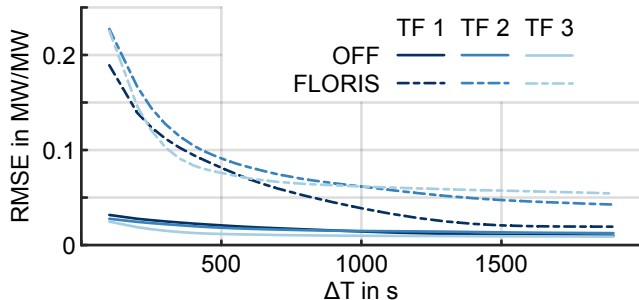

**Figure 16.** Root-mean-squared error of the wind farm efficiency in LES compared to OFF or FLORIS. The wind farm efficiency is defined as the ratio of the energy generated with LuT control divided by the baseline energy integrated over a given time window.

distributions of FLORIS and OFF become more equal. For large $\Delta T$, FLORIS shows a structural underestimation compared to the LES data, where OFF still predicts values along the ideal correlation line. This observation is also quantifiable with the linear regression parameters: As $\Delta T$ lengthens, the linear coefficient approaches 1, and the bias decreases. This trend is visible for both models; however, OFF consistently presents parameters closer to the ideal values.

Figure 16 investigates the error of the approximation of the farm efficiency to further quantify and compare the differences. For each TF and each simulation environment $\eta(t, \Delta T)$ is calculated for $\Delta T \in [100, 1900]$ s and $t \in [t_0, t_1 - T]$, where $t_0$ is the start time of each TF and $t_1$ is the final time. Figure 16 compares how the root-mean-square error between the $\eta(t, T)$ from the LES and the $\eta(t, \Delta T)$ of OFF and FLORIS changes for different $T$. The difference between the LES and FLORIS improves significantly for longer averaging periods, highlighting its design meant for long-term wind farm behavior. On the other hand,

OFF benefits from the addition of wake dynamics and shows a much lower RMSE values compared to FLORIS. However, this advantage becomes smaller as $\Delta T$ grows larger. As a result, a user has to decide if the added computational cost of OFF in comparison to FLORIS justifies the improvement in prediction.

### 4.4 Computational cost

One of the main motivations for dynamic parametric wake models like OFF, or in extend for FLORIDyn or OnWaRDS, is

the low computational cost compared to high-fidelity numerical methods such as LES, for instance. On the other hand, it is evident that the computational cost has to be higher than the cost of the underlying steady-state wake model. Simplified, the computational cost of both OFF and time-marching FLORIS can be expressed as a function of the number of time steps $n_k$, the number of turbines $n_{\mathrm{T}}$, and the number of observation points $n_{\mathrm{OP}}$:

$$\mathcal{O}_{\mathrm{OFF}} = n_k \Big[ \underbrace{\mathcal{O}_{\mathrm{State\ prop.}}(n_{\mathrm{T}}, n_{\mathrm{OP}}) + \mathcal{O}_{\mathrm{TWF}}(n_{\mathrm{T}}, n_{\mathrm{OP}}) + n_{\mathrm{T}} \cdot [\mathcal{O}_{\mathrm{F.\ run}}(n_{\mathrm{T}}) + \mathcal{O}_{\mathrm{F.\ reinit.}}(n_{\mathrm{T}})]}_{\mathcal{O}_{\mathrm{prediction}}} + \dots$$


$$\dots \mathcal{O}_{\mathrm{corr.}}(n_{\mathrm{T}}, n_{\mathrm{OP}}) + \mathcal{O}_{\mathrm{con.}}(n_{\mathrm{T}}) \Big] + \mathcal{O}_{\mathrm{F.\ init.}} + \mathcal{O}_{\mathrm{OFF\ init}}, \tag{14}$$

$$\mathcal{O}_{\mathrm{FLORIS}} = n_k \cdot \mathcal{O}_{\mathrm{F.\ run}}(n_{\mathrm{T}}) + \mathcal{O}_{\mathrm{F.\ init}} + \mathcal{O}_{\mathrm{con.}}(n_{\mathrm{T}}), \tag{15}$$



where $\mathcal{O}_{\text{State prop.}}$ refers to the cost of the state propagation, $\mathcal{O}_{\text{TWF}}$ to the creation of the TWFs, $\mathcal{O}_{\text{F. run}}$ to the cost of one FLORIS evaluation, $\mathcal{O}_{\text{F. reinit.}}/\mathcal{O}_{\text{F. init.}}$ to the FLORIS re-/initialization, $\mathcal{O}_{\text{corr.}}$ to the state correction, and lastly $\mathcal{O}_{\text{con.}}$ to the derivation of the control set-points. This is accompanied by other costs, such as visualization, data storage, memory limitations, etc., which are excluded here.

Performance analysis during the code development has shown that the reoccurring computational costs of $\mathcal{O}_{\text{F. reinit.}}$ can be substantial depending on the implementation. FLORIS was developed with other simulation goals in mind. This leads to costs associated with the reinitialization that are mandatory for some FLORIS applications but could be neglected for purposes of the OFF simulations. Consequently, existing codes similar to OFF have mainly chosen to implement their own wake model. This, in return, limits the capabilities and flexibility of the wake model, which was one of the main motivations for the development of OFF. Another consideration to reduce computational costs is to only run relevant turbines in the steady-state simulation and thereby decrease the cost of $\mathcal{O}_{\text{F. run}}(n_{\text{T}})$. This could be done by excluding turbines that do not contribute to the wake losses experienced by the turbine the TWF is dedicated to. The validity of this approach also depends on the steady-state model capabilities. For instance, if there is a blockage model based on $n_{\text{T}}$, this simplification would introduce a systematic model error. Lastly, parallelization is a natural approach to improving computational complexity. The $n_{\text{T}}$ TWF evaluations done in one time step can be done independently of one another, which would lead to a performance improvement for up to $n_{\text{T}}$ cores. In this work, we investigated a large number of control settings and, therefore, used OFF as a single-core code and split the task at hand over multiple simulations. To give an estimate, in our ten-turbine simulations, OFF ran with a real-time factor of $2.2 \cdot 10^{-1}$ in single-core performance, resulting in 5 h 20 min CPU time for 23 h 45 min simulated time. The SOWFA simulations, recalculated from 80 cores to one core, ran with a real-time factor of $2 \cdot 10^{3}$, resulting in 6030 h CPU time for 3 h simulated time. Lastly, FLORIS ran with a real-time factor of $5.2 \cdot 10^{-5}$, resulting in $4.43$ s wall time for 23 h 45 min simulated time. Previous work showed that the real-time factor of a model like OFF can be reduced to the order of $10^{-3}$ for a similar-sized wind farm with a dedicated implementation of the Gaussian wake model (Lejeune et al., 2022; Becker et al., 2022b).

## 5 Conclusions

This paper introduces OFF, a dynamic open-source wake model designed for wind farm flow control, wake model development, and as a unified interface for various similar models. In this context, a generic description of a passive Lagrangian particle wake model is provided, along with details on the specific version used to achieve the results discussed here. In an example case, the model is used to make an informed parameter choice for a wake steering controller before verifying the selected settings in LES. The controller applies a wake steering look-up table dynamically for a ten-turbine wind farm. The wind farm layout is based on the Hollandse-Kust-Noord wind farm, and the about 24 hour long period of wind direction time series used to test the controllers is based on field data from the same location.

The results from the study show that the wind farm controller can lead to suboptimal performance in the presence of wind direction changes compared to what was predicted during the generation of the LuT based on a steady-state assumption. The



study also shows that the wake steering controller's performance can vary widely for the same wind direction based on the prior state of wind direction, wakes, and used controller. Six selected sets of controller settings are then verified in LES in three 3 hour long subsets of the wind direction change time series. The results show overall good agreement between the LES and OFF in both predicted power generated and wake steering controller efficiency. The LES, for instance, confirms that one of the selected time frames poses a challenging environment for the wake steering controller to return consistent gains over the

baseline operation. The results further investigate the time scales described by both FLORIS and OFF. A conclusion drawn from the comparison is that a dynamic wake description leads to a better correlation with the LES power signal, as well as a lower root-mean-squared error compared to a steady-state prediction.

In conclusion, OFF provides a unified interface to a dynamic wake description that is advantageous over steady-state wake

models for shorter time periods (< 20 min). The model is open-source and designed to interface with steady-state wake model toolboxes. This has been demonstrated with the FLORIS toolbox. As a result, users of OFF can also benefit from the ongoing development done for the underlying wake models.

Future work should further investigate the use and effect of various steady-state wake models in a dynamic context. This

starts with further validation of the approach and the generation of more realistic test and reference cases. It may also involve investigating the selection of wake parameters. Since OFF describes wakes at higher frequencies, the resulting wake shape may appear more slender than a steady-state wake, which must account for small-scale wind direction changes and wake meandering. The OFF code is further built modular to be expanded by other dynamic elements and to further explore their effectiveness for the description of dynamic flows. This includes for instance wake advection descriptions (*e.g.* Zong and Porté-Agel (2020);

Starke et al. (2023)) or floating turbine dynamics (*e.g.* Kheirabadi and Nagamune (2021)). Another direction of interest can be the employment of single-wake dynamic surrogate models in a wind farm, *e.g.* Bastine et al. (2015); Gutknecht et al. (2023).

  In the long term OFF should lead towards a new dynamic wake model that replaces modularity with reduced computational cost and a dedicated, informed selection of the components previously explored.

*Code and data availability.* The OFF framework is available on GitHub under `github.com/TUDelft-DataDrivenControl/OFF`,

the code used for this publication can be found with `doi.org/10.4121/331f86fe-5acb-4a60-99cd-7f8f0135c200` or on the repository with the commit 7910dd2e960821bc85e1468efe24f3cf8b5602cf. The data generated and used in this paper is available with `doi.org/10.4121/29c209fa-f2a4-456d-9353-67cf81be1aaa` on the data.4tu.nl website. It also includes plotting and post processing scripts to give examples how the data can be used.

*Author contributions.* conceptualization MB ML JWvW PC DA RM, methodology MB ML, software MB ML RM, validation MB ML,

investigation MB ML, writing – original draft MB ML, writing – review MB ML JWvW PC RM, editing MB ML, supervision JWvW PC DA, resources JWvW, funding acquisition JWvW PC.



*Competing interests.* At least one of the (co-)authors is a member of the editorial board of Wind Energy Science.

*Acknowledgements.* This work is part of the research programme "Robust closed-loop wake steering for large densely spaced wind farms" with project number 17512, which is (partly) financed by the Dutch Research Council (NWO).

This project has received funding from the European Research Council under the European Union's Horizon 2020 research and innovation program (grant agreement no. 725627).

This work has been supported by the SUDOCO project, which receives the funding from the European Union's Horizon Europe Programme under the grant No. 101122256.





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
