# Peer review of "A dynamic open-source model to investigate wake dynamics in response to wind farm flow control strategies"

_Wind Energy Science, 2024_

## Referee Comment (RC2)

Review – wes-2024-150

**A dynamic open-source model to investigate wake dynamics in response to wind farm flow control strategies**

Marcus Becker, Maxime Lejeune, Philippe Chatelain, Dries Allaerts, Rafael Mudafort, and Jan-Willem van Wingerden

**Review - Anonymous Referee #2**

**1. General comments**

This paper is a very good piece of work on the development of a new wake modeling framework including unsteady effects. It is a highly relevant work for the present literature in the domain. I recommend this paper to be published with minor revisions.

Overall the paper is bit unbalanced between a rather short methodological part (Section 2), and a very large results part (Sections 3 and 4).

Some subsections are quite very large (as 4.2) and could deserve to be split in two separated. At line 346 starts clearly a new sub-topic where this could be divided. Also at 386.

Several figures are using unconventional ways of presenting data, and often too much data was put into them. It takes a lot of time for the reader to understand these figures good.

**2. Specific comments**

- Line 2: I would rather list in the order "like power maximization, power tracking or load mitigation." In order to logically follow historical development of WFFC discipline (which started mostly for power maximization while the later two are more state-of-the-art research).

- Line 4: "OFF": can you already here specify what the acronym stands for? It would help the reader to understand already here that it's a combination framework.

- Line 102-104: I would add again the mathematical notation in each sentence to help clarify these definitions. "Turbine states $x_T$ consist of ... . The ambient states $x_{amb}$ characterize..."

- Line 141: including the vertical deflection (w component) would be useful in future work not only for terrain effects, but also rotor tilting (which is quite common) or floating wind. I would complete here. In fact, it was observed that the absence of vertical deflection in steady state models (e.g. FLORIS) also create discrepancies compared to DWM models with rotor tilt (e.g. FAST.Farm.).

- Line 161-162: It might be a typo or my own misunderstanding, but why should the number of time-steps equals the number of turbines in the farm (both denoted $n_t$)?

- Line 166: "power coefficient $C_p(u)$ and thrust coefficient $C_t(u)$ tables (u being the wind speed ahead)". For non-initiated readers.

- Line 166: "cosine-loss law for yaw misalignment". Add a reference for it.

- Line 226: Can OFF handle veer? If not, it can be cited for future work.

- Line 241: is it due to the choice of the cosine-loss factor? Which factor was used and why? (Limitations of this cosine-loss law have been published in the literature, as the loss factor should actually be varying with ambient conditions such as shear and veer and control set-points of the rotor).

- Figure 7: color legend scale for Phi_lim?

- Figure 7: can it really be called a "Pareto front"? As this does not really result from a multi-objective optimization between energy increase and yaw travel. I don't think these points are really non-dominated.

- Line 256: same comment Pareto front.

- Line 273: "wind farm efficiency predicted by the LuT is indeed an upper limit." But on the Figure 8 (b) one can see that sometimes the simulated efficiency goes higher than the predicted one (between 200 and 220deg). Why does this happen? The above statement should be changed.

- Line 340: What could be a reason(s) for that? How could this be improved in OFF?

- Line 341-342: Is this a (synthetic) smoothing effect that while the power of some turbine is underestimated, the power of others is overestimated? This should be more clearly stated. Furthermore, is this farm-level smoothing expected to be always the case? Maybe in different scenarios, the mismatch of several turbines would add on top of each other for the farm level.

- Line 346: Here I would suggest to start a new subsection (4.3). 4.2 is overall quite large already, splitting in two can be good. At this line a new (sub-) research question is starting.

- Line 361-363: please make uniform the two results presentation and units (one writes f_cutoff the other one no, one expresses in s^-1 the other one in Hz, one gives the full final value in 0.0027Hz the other one no).

- Line 361-363: 0.0027Hz and 0.0019Hz. How can these frequencies be physically interpreted? To me it is a bit hard to link back to real physics of the flow (very low frequencies no?). As these two results are a main core results of the whole paper (already cited in the abstract), I think it would be great to explain them more and make a link with the physical world. I feel a bit frustrated to not manage to grasp it now.

- Figure 13 (a): please add legend (one should not need to read the caption to see the meaning of the colors). The different lines are for different turbines? Also a bit unclear.

- Figure 14: The figure is a bit messy and unclear. Unconventional way of showing data. It takes time for the reader to grasp the meaning of it.

- Line 381-382: Yes, this is the cosine-loss law correction that should definitely be included for future work to improve this issue. This crucial point should already have been mentioned above also (see comment on the limitation of the cosine-loss above). See also: Tamaro et al. 2024 https://doi.org/10.5194/wes-9-1547-2024

- Line 386: here a new subsection could be started.

- Sections 4.3 and 4.4 are very well written and clear (more than 4.2 that could be clarified).

---

## Author Comment (AC1)

**Response to the reviewers**

Publication *A dynamic open-source model to investigate wake dynamics in response to wind farm flow control strategies*

Dear reviewers,

Thank you for the kind words regarding the publication and thank you for the time invested in the review and the valuable feedback. We have sorted the comments based on the structure of the paper and distinguished between reviewer 1 (**R1**) and reviewer 2 (**R2**). Our responses are split into a response comment (**C**) to the reviewer and a summary of the modifications (**M**) made to the manuscript.

With kind regards,

Marcus Becker
Maxime Lejeune

**Abstract**

**R2** Line 2 *I would rather list in the order "like power maximization, power tracking or load mitigation."* *In order to logically follow historical development of WFFC discipline (which started mostly for power maximization while the later two are more state-of-the-art research).*
**M** Adapted the proposed order.

**R2** Line 4 *"OFF": can you already here specify what the acronym stands for? It would help the reader to understand already here that it's a combination framework.*
**M** Changed

> *This paper presents an open-source wake modeling framework called OFF.*

to

> *This paper presents an open-source wake modeling framework called OFF (abbreviated from the models OnWARDS, FLORIDyn and FLORIS).*

**R1** *No comments*

**Section 1 Introduction**

**R1** *No comments*
**R2** *No comments*

**Section 2 Model description**

**R2** Line 102-104 *I would add again the mathematical notation in each sentence to help clarify these definitions. "Turbine states $x\_T$ consist of … . The ambient states $x\_{amb}$ characterize…"*
**M** The proposed changes have been implemented.

**R2** Line 141 *Including the vertical deflection (w component) would be useful in future work not only for terrain effects, but also rotor tilting (which is quite common) or floating wind. I would complete here. In fact, it was observed that the absence of vertical deflection in steady state models (e.g. FLORIS) also create discrepancies compared to DWM models with rotor tilt (e.g. FAST.Farm.).*
**R1** Line 142 *Use some more words to describe "no tilting" (e.g. "no or small tilt angles on the wind turbine rotors", or "no rotor-tilt-based wake redirection").*
**C** Tilt and yaw steering is an interesting case where the border between FLORIDyn (and thus OFF) and FLORIS really depends on the implementation. In essence, FLORIDyn uses the OP states to retrieve the resulting wind speed reduction from an arbitrary wake model, FLORIS, in this case. If the turbine states result in a deflection of the wake, the OPs don't necessarily have to follow this deflection; only the wake shape in the wake model does. The upside of this design is that as soon as models like FLORIS support tilt steering (which it does https://nrel.github.io/floris/examples/examples_floating/003_tilt_driven_vertical_wake_deflection.html) FLORIDyn and OFF also support it. The downside is that the OPs no longer represent the centerline, rather just a chain of passive tracers behind the rotor.
The resulting formulation of the state propagation is, therefore, also slightly different from previous formulations, as it does not include the deflection term from the wake model.
**M** This discussion is not in the current version of the manuscript, and changes have been made to include it and to clarify the design:

> *Note that similar, more detailed state-space descriptions can be found with Gebraad and van Wingerden (2014); Becker et al. (2022a); Foloppe et al. (2022). The code internally decomposes the wind speed and direction into its u and v components to avoid unexpected behavior when switching between 360 and 0 deg. These are then used along with the time step $\Delta t$ to advance the location of the OPs through a Lagrangian update; see Equation (6). The w component is ignored for simplicity. Accounting for the vertical deflection of the wake center might become necessary in some contexts, e.g. for simulations including terrain. However, it was not deemed necessary for the application presented here, i.e., an offshore wind farm with no tilting.*

Was changed to

> *Note that similar, more detailed state-space descriptions can be found with Gebraad and van Wingerden (2014); Becker et al. (2022a); Foloppe et al. (2022).* *A difference between these formulations and the one employed in OFF is that OFF's formulation does not include vertical or horizontal OP deflection based on the yaw and tilt angle of the turbine. Rather, the impact of yaw and tilt turbine misalignment on the wake shape is solely simulated in the wake model. The code internally decomposes the wind speed and direction into its u and v components to avoid unexpected behavior when switching between 360 and 0 deg. These are then used along with the time step $\Delta t$ to advance the location of the OPs through a Lagrangian update; see Equation (6). The w component is ignored for simplicity. Accounting for the vertical deflection of the wake center might become necessary in some contexts, e.g. for simulations including terrain. However, it was not deemed necessary for the application presented here, i.e., an offshore wind farm* .

**R2** Line 161-162 *It might be a typo or my own misunderstanding, but why should the number of time-steps equals the number of turbines in the farm (both denoted $n_t$)?*

**C** The text aims to say that for each turbine, a dedicated steady-state model simulation is necessary. Each dedicated steady-state simulation contains locations for all turbines in the wind farm. Therefore at each time step, each turbine needs to run a steady-state simulation with all turbines.

**M** The text was changed from

> *At each time step, a new TWF is generated for each turbine individually, which leads to nT simulations of nT turbines.*

To

> *At each time step, a new individual TWF is generated for each of the nT turbines. This leads to nT TWF simulations, where each of them contains nT turbines.*

**R1** Line 165 *Avoid a vague use of language 'mirror a possible "out of the box" experience'*

**M** The sentence was rephrased from

> *No parameter tuning was performed to mirror a possible "out of the box" experience.*

To

> *No parameter tuning was performed to represent the performance achievable with the default settings.*

**R2** Line 166 *"power coefficient Cp(u) and thrust coefficient Ct(u) tables (u being the wind speed ahead)". For non-initiated readers.*

**M** The text was adapted accordingly.

**R2** Line 166 *"cosine-loss law for yaw misalignment". Add a reference for it.*

**M** The text was modified from

> *The turbine model within FLORIS is based on the c_p(u) and c_t(u) tables (u being the wind speed ahead) of the DTU 10 MW (Bak et al.), corrected with the cosine-loss law for yaw misalignment.*

To

> *The turbine model within FLORIS is based on the c_p(u) and c_t(u) tables (u being the wind speed ahead) of the DTU 10 MW (Bak et al.), corrected with the blade element momentum theory based cosine-loss law for yaw misalignment (Rankine, 1865; Froude, 1889). Specifically, the classical value of 1.88 is retained for the cosine power-loss law exponent. We nonetheless acknowledge that this constant power-loss model does not account for the variability of operating conditions and will therefore likely affect the optimal steering angles computed, as noted by Tamaro et al. (2024).*

**R1** Figure 2 *Add a label "ghost OP" to the relevant element in Figure 2*

**M** The Figure has been adapted accordingly.

**R1** Section 2.3 *First it is stated that wind direction is used to evaluate the LUT, then later TI, Wind Speed and Wind Direction. Could be explained more clearly.*
**M** The text was modified from

> *The primary input parameters for the LuT are derived from the freestream atmospheric conditions, which are parameterized as hub-height Turbulence intensity (TI), wind speed, and wind direction. While the TI is kept constant at 6%, the wind direction is discretized into 1 deg bins, and the wind speed from 6 ms⁻¹ to 10 ms⁻¹ in 1 ms⁻¹ steps.*

To

> *While the presented control law focuses on wind direction changes, for completeness, the provided lookup table (LuT) also includes inputs for other freestream atmospheric conditions, such as hub-height Turbulence intensity (TI) and free wind speed. These parameters are kept constant in the case study discussed in Section 3. During the LuT creation, TI is kept constant at 6%, the wind direction is discretized into 1 deg bins, and the wind speed from 6 ms⁻¹ to 10 ms⁻¹ in 1 ms⁻¹ steps.*

**R1** Section 2.3 *List clearly some relevant parameters of the controller, namely the update time of the controller, and the maximum yaw amplitude. I was not able to find them.*
**C** Thank you for spotting this short coming of the description.
**M** Section 2.3 has been extended by the sentence

> *The controllers are continuously updated with every 5 s time step of the simulation; the limits of the intentional misalignment with the main wind direction are set to ±30 deg.*

**Section 3 Simulation setup**

**R2** Line 226 *Can OFF handle veer? If not, it can be cited for future work.*
**C** In a (crude) way, it does: If the surrogate wake model changes the wake shape due to veer, OFF will show the same behavior. However, OFF does not support the advection of different parts of the wake at different speeds. I can imagine that this does become more relevant in atmospheric conditions with strong veer, where the layers of air mix less and advect with different speeds and directions.
**M** The handling of sheared and veered conditions was mention in the conclusion.

> *The OFF code is further built modular to be expanded by other dynamic elements and to further explore their effectiveness for the description of dynamic flows. This includes for instance wake advection descriptions (e.g. Zong and Porté-Agel (2020); 490 Starke et al. (2023)) or floating turbine dynamics (e.g. Kheirabadi and Nagamune (2021)).*

To

> *The OFF code is further built modular to be expanded by other dynamic elements and to further explore their effectiveness for the description of dynamic flows. This includes for instance wake advection descriptions (e.g. Zong and Porté-Agel (2020); Starke et al. (2023); Lejeune et al. (2022)), shear and veer parametrizations (e.g. Abkar et al. (2018)) or floating turbine dynamics (e.g. Kheirabadi and Nagamune (2021))*

**R2** Line 241 *Is it due to the choice of the cosine-loss factor? Which factor was used and why? (Limitations of this cosine-loss law have been published in the literature, as the loss factor should actually be varying with ambient conditions such as shear and veer and control set-points of the rotor).*
**C** The power-loss parametrization used is the classical $\cos^p \gamma$ with $p = 1.88$. This corresponds to the default Floris used by FLORIS. We acknowledge the limitations of this formulation which may reflect into improper computation of the optimal steering angles. Yet, we chose to keep the default value of the constant power-loss law coefficient as evaluating the sensitivity of our approach to this parameter was out of scope of the present paper and as this value remains widely used despite its limitations.
**M** In addition to the more detailed description of the power-loss model used (Section 6, comment 6). The text was updated from

*This is a result of a power curve that has little sensitivity to small yaw angle misalignments.*

To

*This is a result of a power curve that has little sensitivity to small yaw angle misalignment, possibly highlighting the need for more adequate power-loss exponent parametrization (Tamaro et al., 2024).*

**R2** Figure 7 *Color legend scale for Phi_lim?*
M Color legend has been added.

**R2** Figure 7 *Can it really be called a "Pareto front"? As this does not really result from a multi-objective optimization between energy increase and yaw travel. I don't think these points are really non-dominated.*
**R2** Line 256 *Same comment Pareto front.*
**C** To some extent, this figure can be described as the result of a multi-objective optimization using a grid-search algorithm. Given the defined search space, all solutions considered part of the Pareto front are indeed non-dominated. However, we acknowledge that a more precise term would be „approximated Pareto front", as a refined optimization could shift its boundaries
**M** Renamed pareto front to „approximated Pareto front"

**R2** Line 273 *"wind farm efficiency predicted by the LuT is indeed an upper limit." But on the Figure 8 (b) one can see that sometimes the simulated efficiency goes higher than the predicted one (between 200 and 220deg). Why does this happen? The above statement should be changed.*
**C** The wind farm efficiency predicted by the LuT provides an upper limit to the potential gains achievable under steady-state conditions (Lejeune et al., 2024). The occasional, localized overshoots beyond this performance envelope can be attributed to the dynamic nature of the simulations. For instance, in the absence of wake steering, a downstream turbine aligned with the wind direction would always operate within the wake of the upstream turbine. However, in a dynamic setup, transient wind direction changes may temporarily shift the wake, allowing the downstream turbine to operate under improved conditions and produce more power than in the steady-state scenario. Nevertheless, these overshoots are temporary, eventually converging back to the steady-state value or lower. This observation precisely highlights the need for dynamic wake models that can optimize wind farm control strategies during transient periods.
**M** The text was modified to include this discussion:
The difference between the achieved wind farm efficiency and the predicted one goes to show that the changing turbine states and wind direction state can lead to suboptimal performance and that the wind farm efficiency predicted by the LuT is, in most cases, an upper limit, only achievable under steady-state conditions (Lejeune et al., 2024). The occasional localized overshoots beyond this performance envelope can be attributed to the dynamic nature of the simulations. For instance, in the absence of wake steering, a downstream turbine aligned with the wind direction would always operate within the wake of the upstream turbine. However, in a dynamic setup, transient wind direction changes may temporarily shift the wake, allowing the downstream turbine to operate under improved conditions and produce more power than in the steady-state scenario. Nevertheless, these overshoots are temporary, eventually converging back to the steady-state value or lower. This observation highlights the need for dynamic wake models that can optimize wind farm control strategies during transient periods.

**R1** *No comments*

**Section 4 HKN Cases simulated in LES**

**R2** Line 340 *What could be a reason(s) for that? How could this be improved in OFF?*
**C** The most likely reason is that OFF is essentially a middle ground between FLORIS and the LES, describing some of the flow dynamics but ignoring most of the stochasticity of the flow. FLORIS provides an upper bound to the gains achievable by wake steering but, in practice, these gains are

never quite reached (Lejeune et al., 2024). OFF, as a middle ground between FLORIS and LES, thus still inflates the gains obtained in the LES but to a lesser extent than FLORIS.

**M** The manuscript was extended by the following comment:

OFF tends to either match or overestimate the effect of yaw steering on the turbine efficiency, compared to the filtered LES signal. This may be attributed to the fact that OFF describes a middle ground between an over-confident steady-state model and a more realistic LES simulation.

**R2** Line 341-342 *Is this a (synthetic) smoothing effect that while the power of some turbine is underestimated, the power of others is overestimated? This should be more clearly stated. Furthermore, is this farm-level smoothing expected to be always the case? Maybe in different scenarios, the mismatch of several turbines would add on top of each other for the farm level.*

**C** The improved relative accuracy at the wind farm level is indeed a consequence of errors averaging out when summed over the farm. However, certain scenarios may lead to an increased power production bias at the farm scale:

- **Wake interactions**: Errors may accumulate due to an improper selection of wake expansion constants and/or wake superposition models. While numerous studies have explored these effects in details (eg. (Gunn et al., 2016; Zong & Porté-Agel, 2020)), this issue arises from limitations of the underlying steady-state wake model rather than the dynamic framework presented here. It is therefore not investigated here.
- **Wake advection**: Advecting wakes at a non-physical velocity could further degrade power predictions at the wind farm scale. However, this can be prevented by ensuring an appropriate selection of the wake model parameters. Dynamic wake advection is generally expected to improve agreement with LES. Even if wake advection model is quite crude, it is still more faithful to the underlying flow physics than ignoring advection entirely.
- **Power curve:** Efficiency is a fraction of the controlled performance with respect to the baseline. A waked turbine is expected to generate less power overall in the first place, adding control can thereby significantly increase the effective rotor wind speed and, by the power of three, the power of the turbine. Across a full farm however, freestream turbines generate the majority of the power, making the contributions of waked turbines less significant.

**M** added the statement:

The improved performance on a farm scale may stem from different sources: (i) The fact that turbines are distributed throughout the farm makes it more likely that if one is not waked, another one may be. As a result under- and overestimation may cancel out. (ii) Looking at an individual turbine, small increases in wind speed lead to a large amplification of the power generated. As a result, mismatches create a large error. However, in the presented farm context the power contribution of waked turbines is small compared to the free-stream turbines.

**R2** Line 346 *Here I would suggest to start a new subsection (4.3). 4.2 is overall quite large already, splitting in two can be good. At this line a new (sub-) research question is starting.*

**C** Has been addressed, see *General comments*.

**R2** Line 361-363 *Please make uniform the two results presentation and units (one writes f_cutoff the other one no, one expresses in s^-1 the other one in Hz, one gives the full final value in 0.0027Hz the other one no).*

**M** The text has been adapted for consistency.

**R2** Line 361-363 *0.0027Hz and 0.0019Hz. How can these frequencies be physically interpreted? To me it is a bit hard to link back to real physics of the flow (very low frequencies no?). As these two results are a main core results of the whole paper (already cited in the abstract), I think it would be great to explain them more and make a link with the physical world. I feel a bit frustrated to not manage to grasp it now.*

**C** These frequencies should be interpreted in terms of the rotor-based convective time scale $D/U$:

- $f_{\text{FLORIS}}$ corresponds to $(D/U)/23.6$ ;

- $f_{\text{OFF}}$ corresponds to $(D/U)/16.8$

The typical time scale for wake meandering spans $1/(20\,U/D) < f_m < 1/(2U/D$ to (Larsen et al., 2007; Lio et al., 2021; Onnen et al., 2025). While the characteristic temporal frequency described by FLORIS falls well outside this range, we observe that OFF, with its improved dynamics, is able to capture some of the wake dynamics within this time range.

However, capturing the full spectrum of scales associated with wake meandering is beyond the scope of the current OFF framework as presented here. Nevertheless, OFF is built modular, allowing for experimentation with alternative parametrization of the wake propagation. For example, one could implement advection schemes similar to (Lejeune, 2023), which offer a more detailed description of wake advection, notably capturing time scales relevant to wake meandering.

Additionally, we note that, following (Van Den Broek et al., 2024), a low-pass filter with a cutoff frequency of 1/600 Hz or $1/(5.38t_c)$ was applied to the wind direction retrieved from the LiDAR measurements. The filtered output was then used as input for both the yaw steering controllers and the wind direction precursor.

We have a posteriori knowledge that this cutoff frequency may, therefore, be too low to demonstrate the full capabilities of OFF. However, rerunning simulations with a higher cutoff filtering frequency was deemed too computationally expensive within the scope of this work.

**M** An addition was made to the manuscript which puts the frequencies in context of wake meandering frequencies:

This cutoff also aligns with the literature on wake meandering, which is not captured by OFF:

Lio et al. (2021) finds the wake meandering frequency to be around u∞/20D , which equals 0.0022 Hz for the presented study. Larsen et al. (2007) on the other hand suggests a higher frequency, which, for this study, equals 0.022 Hz. We can conclude that OFF does describe the wake dynamics up to the wake meandering frequency.

And a second addition, which suggests where to filter instead:

It should be noted that the filtering timescale used to preprocess the wind direction signal (1/600 Hz) may limit OFF's performance, as it filters out relevant dynamic scales. Related work by (Simley et al., 2020) suggests, for instance, that mean wind direction changes may occur with a frequency of up to 1/270 Hz. Rerunning the LES with a higher cutoff frequency would likely increase OFF's effective cutoff frequency estimation; however, this was not feasible within the scope of the present work.

**R2** Figure 13 (a) *please add legend (one should not need to read the caption to see the meaning of the colors). The different lines are for different turbines? Also a bit unclear*
**M** Made changes to the caption, axis label and legend.

**R2** Figure 14 *The figure is a bit messy and unclear. Unconventional way of showing data. It takes time for the reader to grasp the meaning of it.*
**C** We agree that the graph is complex but also feel it provides more depth to the results depicted in Figure 13 (b). In essence, it shows where the minimums presented in 13 (b) stem from and may give pointers to where the underlying wake model, and/or the dynamic description requires improvements. Given that Figure 13 (b) depicts one of the main results of the paper, we felt that Figure 14 is a valuable addition for future developments.

**R2** Line 381-382 *Yes, this is the cosine-loss law correction that should definitely be included for future work to improve this issue. This crucial point should already have been mentioned above also (see comment on the limitation of the cosine-loss above). See also: Tamaro et al. 2024 https://doi.org/10.5194/wes-9-1547-2024*
**M** Added source, note that this suggestion was also taken into account with an earlier comment. (Line 251 in the revised manuscript)

**R2** Line 386 *Here, a new subsection could be started.*

**C** Addressed, see *General comments*

**R1** Section 4.3 *Already mention the 20 minutes in Section 4.3, presumably considered the threshold for "large ΔT", which later comes back in the Conclusions (line 480)*
**C**
**M** A statement was added to the end of Section 4.5 (former 4.3):
We conclude that, based on this case study, it is advantageous to use OFF for quantities of interest shorter than ≈ 20 min. However, for longer time scales the benefit of the added dynamics diminishes.

**Section 5 Conclusions**
R1: *No comments*
R2: *No comments*

**General comments**

**R1** *Further, what is less convincing on the modeling, is that in the case study only wind direction is varied. It would be good at this point to refer specifically to other studies with variation of wind speed as well.*
**C** The limitation to the wind direction changes is discussed in Section 3.1. One reason is that the used field data may be influenced by neighboring wind farms. However, the limitation also lowers the complexity of the controller, which would otherwise need to take operation in Regions 1 and 3 into account, as well as the transitions. A suitable design of such a controller is outside the paper's scope. Following your suggestion, we adapted the conclusions of the manuscript to link to two related publications that focus on reduced order model performance during wind speed changes while applying control.
**M** Future work should further investigate the use and effect of various steady-state wake models in a dynamic context. This starts with further validation of the approach and the generation of more realistic test and reference cases. One shortcoming of the presented case study is its limitation to wind direction variations. Future work should investigate the model and control performance with realistic wind speed variations, similar to the works of e.g. (Doekemeijer et al., 2020; Van Den Broek et al., 2024)

**R2** *Overall the paper is bit unbalanced between a rather short methodological part (Section 2), and a very large results part (Sections 3 and 4).*
**R2** *Some subsections are quite very large (as 4.2) and could deserve to be split in two separated. At line 346 starts clearly a new sub-topic where this could be divided. Also at 386.*
**C** The stronger emphasis on the case study and comparison is by design, as we feel previously published papers already dove into the mathematics of the implemented model. Therefore, we feel that the paper's main contribution is a showcase and in-depth discussion of a case study.
We do agree with the observation that Section 4.2 is too long and followed the suggestion to split it into three sections.
**M** Section 4.2 was split into „4.2 Power generated", „4.3 Power signal correlation", and „4.4 Power error statistics" , The introduction of Section 4 was adapted accordingly

**R2** *Several figures are using unconventional ways of presenting data, and often too much data was put into them. It takes a lot of time for the reader to understand these figures good.*
**C** Thank you for the criticism and feedback. We know that some of the figures in the publication may be unconventional. The idea was to add new perspectives on the data alongside more familiar plots. Additionally, we wanted to present a comprehensive view of the model's performance across multiple hours of data and a large selection of controllers. Still, we also wanted to refrain from solely reducing it to statistics. As a result, we had to select some data to give an idea of the overall performance and feel that the now given information draws a more complete picture of what a software user might experience.

**Bibliography**

Doekemeijer, B. M., Van Der Hoek, D., & Van Wingerden, J.-W. (2020). Closed-loop model-based wind farm control using FLORIS under time-varying inflow conditions. *Renewable Energy*, *156*, 719–730. https://doi.org/10.1016/j.renene.2020.04.007

Gunn, K., Stock-Williams, C., Burke, M., Willden, R., Vogel, C., Hunter, W., Stallard, T., Robinson, N., & Schmidt, S. R. (2016). Limitations to the validity of single wake superposition in wind farm yield assessment. *Journal of Physics: Conference Series*, *749*(1), 012003. https://doi.org/10.1088/1742-6596/749/1/012003

Larsen, G. Chr., Madsen Aagaard, H., Bingöl, F., Mann, J., Ott, S., Sørensen, J. N., Okulov, V., Troldborg, N., Nielsen, N. M., Thomsen, K., Larsen, T. J., & Mikkelsen, R. (2007). *Dynamic wake meandering modeling*. Risø National Laboratory.

Lejeune, M. (2023). *A meandering-capturing wake model coupled to rotor-based flow-sensing for operational wind farm flow estimation* [PhD Thesis]. UCLouvain.

Lejeune, M., Frère, A., Moens, M., & Chatelain, P. (2024). Are steady-state wake models and lookup tables sufficient to design profitable wake steering strategies? A Large Eddy Simulation investigation. *Journal of Physics: Conference Series*, *2767*(9), 092075. https://doi.org/10.1088/1742-6596/2767/9/092075

Lio, W. H., Larsen, G. Chr., & Thorsen, G. R. (2021). Dynamic wake tracking using a cost-effective LiDAR and Kalman filtering: Design, simulation and full-scale validation. *Renewable Energy*, *172*, 1073–1086. https://doi.org/10.1016/j.renene.2021.03.081

Onnen, D., Larsen, G. C., Lio, A. W. H., Hulsman, P., Kühn, M., & Petrović, V. (2025). *Field comparison of load-based wind turbine wake tracking with a scanning lidar reference*. Dynamics and control/Wind farm control. https://doi.org/10.5194/wes-2024-188

Simley, E., Fleming, P., & King, J. (2020). Design and analysis of a wake steering controller with wind direction variability. *Wind Energy Science*, *5*(2), 451–468. https://doi.org/10.5194/wes-5-451-2020

Van Den Broek, M. J., Becker, M., Sanderse, B., & Van Wingerden, J.-W. (2024). Dynamic wind farm flow control using free-vortex wake models. *Wind Energy Science*, *9*(3), 721–740. https://doi.org/10.5194/wes-9-721-2024

Zong, H., & Porté-Agel, F. (2020). A momentum-conserving wake superposition method for wind farm power prediction. *Journal of Fluid Mechanics*, *889*, A8.